# Degraders of the dengue virus capsid protein exhibit differentiated pharmacology relative to capsid inhibitors

Antara Chakravarty [1,6], Lu-Ning Wang [1,6], Ryan P. Golden [2], Zhengnian Li [2,5], Katherine A. Donovan [3,4], Oshri Afanzar[1], Yupeng Zhang[1], Eric S. Fischer [3,4], Nathanael S. Gray [2] & Priscilla L. Yang [1]✉

Due to the limited size of viral genomes, most viral proteins are multi-functional; yet most direct-acting antivirals are designed as single-function inhibitors. The dengue virus (DENV) capsid protein serves as a building block for new virions while also interacting with multiple host factors to remodel the cellular environment. Using established capsid inhibitor ST148 as a targeting ligand, we develop a DENV capsid degrader, RPG-01-132, that exhibits a broadened spectrum of activity against the four DENV serotypes and an ST148-resistant mutant virus. Using multiple approaches, we show that RPG-01-132's sub-micromolar antiviral activity is due to CRL4$^{CRBN}$-dependent degradation of capsid and that this mechanism disrupts capsid-related pathways required for productive infection, including infectious virus output and capsid-mediated antagonism of the interferon response. This pharmacology is well-differentiated from ST148, which interferes with assembly of new virions, but has no demonstrated effect on the capsid's nonstructural functions. These findings demonstrate that targeted protein degradation can thus enable antiviral pharmacology not observed with conventional antiviral inhibitors and that is resilient to point mutations that reduce inhibitor potency.

Heterobivalent small-molecule degraders, also known as PROTACs (PROteolysis TArgeting Chimeras), induce targeted protein degradation (TPD) through the formation of a ternary complex with the target protein and an E3 ubiquitin ligase[1,2]. Unlike conventional inhibitors requiring continuous occupancy to block enzymatic activity, degraders can exert potent effects with sub-stoichiometric target-engagement due to event-driven pharmacology[3,4]. Interest in the application of TPD as an antiviral strategy has grown in recent years due to its potential advantages. First, the capacity to achieve potent antiviral activity with relatively modest target affinity may enable direct-acting antivirals with broader spectrum activity and higher barriers to resistance compared to typical antiviral inhibitors[5,6]. Second, TPD may provide a strategy to address viral proteins with multiple functions and/or functions that cannot be easily inhibited[7]. While there has been growing success in developing degraders against both enzymatic and non-enzymatic viral targets[7–16] and host proteins[17–21], the discovery of antiviral degraders remains a largely empirical pursuit. Although we at present cannot rationally predict which viral proteins are good targets for a TPD approach, key considerations include the abundance of the target protein, its function(s) in the viral lifecycle, its subcellular localization and accessibility to the ubiquitin-proteasome system (UPS), and the degree of depletion required to see an antiviral effect.

[1]Department of Microbiology and Immunology, School of Medicine, Stanford University, Stanford, CA, USA. [2]Department of Chemical and Systems Biology, ChEM-H, Stanford Cancer Institute, School of Medicine, Stanford University, Stanford, CA, USA. [3]Department of Biological Chemistry and Molecular Pharmacology, Harvard Medical School, Boston, MA, USA. [4]Department of Cancer Biology, Dana-Farber Cancer Institute, Boston, MA, USA. [5]Present address: Septerna, Inc. 250 E Grand Ave., South San Francisco, CA, USA. [6]These authors contributed equally: Antara Chakravarty, Lu-Ning Wang. ✉e-mail: ply@stanford.edu

For instance, viral proteins sequestered in membrane-bound compartments or tightly integrated into large complexes may be less susceptible to TPD mechanisms. Similarly, the degree of depletion required for antiviral efficacy can vary depending on the function of the viral protein, with some targets requiring near-complete depletion and others responding to partial reductions.

Capsid proteins present interesting but potentially challenging targets for TPD. As multifunctional proteins, they are a structural component of virions but also act as effectors to remodel the cellular environment through interactions with a variety of host factors[22–25]. Their lack of human homologs and high degree of conservation are considered advantageous for drug development. Direct-acting antivirals (DAAs) targeting the capsid proteins of both enveloped[26–28] and nonenveloped[29,30] viruses have been approved as drugs or advanced to clinical trials, including small molecules targeting HIV (lenacapavir), hepatitis B virus (ABI-4334, EDP-514, JNJ-56136379, ALG-000184), and enteroviruses (pleconaril, pocapavir). These examples validate capsid proteins as a target class, but their suitability for TPD remains unknown. Due to their primary function as structural components required for the production of new virions, they are expressed in relatively high abundance compared to viral enzymes. Although they are cytosolic and hence accessible to the ubiquitin-proteasome system, their self-assembly into metastable, macromolecular structures may render them challenging to degrade.

Dengue virus (DENV), a human pathogen of the Flavivirus genus transmitted by mosquitoes, is a significant global health threat, causing an estimated 100-400 million infections annually. While many infections are self-limiting, a substantial number progress to severe disease, including dengue hemorrhagic fever and dengue shock syndrome. Due to ongoing concerns regarding the safety of approved vaccines Dengvaxia and TAK-003 in seronegative individuals[31], DAAs that can be used prophylactically or therapeutically remain an unmet biomedical need. The DENV capsid protein (DENV C) has been the focus of both drug discovery efforts and mechanistic studies to define its functions[32–36]. In the cytoplasm, DENV C encapsidates the viral RNA genome to form a nucleocapsid, driving virion assembly through its interactions with endoplasmic reticulum membranes and lipid droplets. Beyond this structural function, DENV C has been shown to remodel the cellular environment through interaction with numerous host proteins, including nucleolin, core histones, and the peroxisomal chaperone PEX19[25,37,38]. In particular, the capsid proteins of DENV and multiple other flaviviruses have been shown to suppress type I interferon responses[25,39]. Pharmacological inhibition of these nonstructural functions is challenging, not least because the mechanisms underlying these functions are not entirely understood. Here, we report an investigation of DENV C as a target for TPD-based DAA development (Fig. 1a), leading to the development of degrader RPG-01-132, which reduces DENV replication in cell culture through specific, TPD-based depletion of DENV C. Compared to ST148[40], the DENV C inhibitor from which it was derived, RPG-01-132 exhibits more consistent antiviral activity across all four DENV serotypes and retains activity against an ST148-resistant DENV2 mutant. RPG-01-132's depletion of intracellular C prevents the formation of virions and mitigates repression of the IFN-β promoter by DENV C. This differs from the mechanism of ST148 and provides proof-of-concept for using TPD strategies against multifunctional viral proteins, including those with both structural and nonstructural functions. Taken together, these experiments validate DENV C as a target for TPD and provide a foundation for the development of TPD-based DAAs targeting this multifunctional viral protein.

## Results

### Design and synthesis of candidate degraders targeting the dengue virus (DENV) C protein

To generate degraders targeting the DENV C protein, we employed ST148, an established DAA targeting DENV C that binds to the C dimer, with two molecules of ST148 stabilizing the formation of C tetramers[40,41]. This impairs normal virion assembly and egress, and ST148 incorporated into progeny virions prevents disassembly of the nucleocapsid during the next cycle of infection. We generated ST148-based degraders employing the phenyl group on ST148 as the exit vector for attachment of the linker based on its exposure in the C-dimer-ST148 complex (PDB 6VG5)[41] (Fig. 1b). Using the *para* position on the phenyl ring of ST148 for attachment, we synthesized *para*-bromo-substituted ST148 as starting material, then employed a palladium-catalyzed Sonogashira coupling reaction of ST148-Br with a propargyl-terminated linker (Supplementary Fig. 1). This was followed by amide condensation to connect the ST148-linker to E3 ligase ligands for both CUL4-RBX1-DDB1-CRBN (CRL4$^{CRBN}$) and CUL2-RBX1-ElonginB-ElonginC-VHL (CRL2$^{VHL}$) (Fig. 1c) to generate ST148-based bivalent compounds (Fig. 1d–f).

### Screening to identify degraders targeting the DENV C protein

To evaluate the activity of these candidate degraders, we systematically assessed their effect on C abundance in DENV2-infected cells. Huh7.5 cells were infected with DENV2 New Guinea C (NGC) at a multiplicity of infection (MOI) of 0.5 for 1 h, washed to remove excess inoculum, and then incubated in the presence of the parental inhibitor ST148 or a candidate degrader at concentrations of 5 µM and 10 µM. Cell lysates were harvested at 24 h post-infection to examine the intracellular abundance of C by Western blot (Fig. 2a and Supplementary Fig. 2a–e). Interpretation of the Western blot data was complicated by ST148's known stabilization of C tetramers and by the potential for a reduction of all viral proteins caused by inhibition of viral replication over the 24-h timeframe of the experiment. To ensure that we could detect any reduction in C caused by the intended TPD mechanism, we included parallel samples in which the free E3 ligase ligand was added as a competitor to block TPD activity. While we also evaluated effects of candidate degraders on viral titers (Supplementary Fig. 2f–j), we ultimately found monitoring C abundance to be the more efficient assay for distinguishing TPD activity from the activity of parental ligand ST148. We initiated our study by examining the degraders with derivatives of thalidomide, a well-known E3 ligand of CRL4$^{CRBN}$, and polyethylene glycol (PEG) linkers. Although the degraders within this set with short (PEG2, RPG-01-138) and long (PEG5, LNW-01-108) linker lengths did not exhibit significant C depletion, compounds RPG-01-139 (PEG3-CRBN) and RPG-01-132 (PEG4-CRBN) with intermediate length linkers caused a large, concentration-dependent decrease in C protein that was blocked in the presence of competing concentrations of the CRL4$^{CRBN}$ ligand lenalidomide (Fig. 2b), consistent with on-mechanism activity. We then synthesized LNW-01-100 (11 methylene units-CRBN), which has a hydrophobic methylene linker of comparable length to RPG-01-132 (PEG4-CRBN). This modification resulted in the loss of degrader activity, with intracellular C either not changing or increasing in a manner that was unaffected by lenalidomide. LNW-01-148, which pairs the PEG4 linker of RPG-01-132 with a structurally distinct CRBN ligand, E3-3[42], also caused no change in C abundance, indicating the importance of the E3 ligand for degrader activity. We additionally synthesized several degraders with PEG linkers and the AHPC-type ligand to recruit CRL2$^{VHL}$. While RPG-01-136 (PEG2) exhibited VHL-dependent activity, increasing the linker by an additional PEG unit (RPG-01-133, PEG3) resulted in reduced activity that was not competed with the VHL ligand, and the PEG4- and VHL-based degrader (RPG-01-137) had no activity.

To more rigorously examine if RPG-01-139 (PEG3-CRBN) and RPG-01-132 (PEG4-CRBN) reduce intracellular C via a CRL4$^{CRBN}$-dependent TPD mechanism, we pursued several complementary approaches. First, we confirmed that RPG-01-139 and RPG-01-132 can engage both C and CRL4$^{CRBN}$ in cells. Second, we investigated whether the reduction of C caused by RPG-01-139 and RPG-01-132 is absent in cells genetically

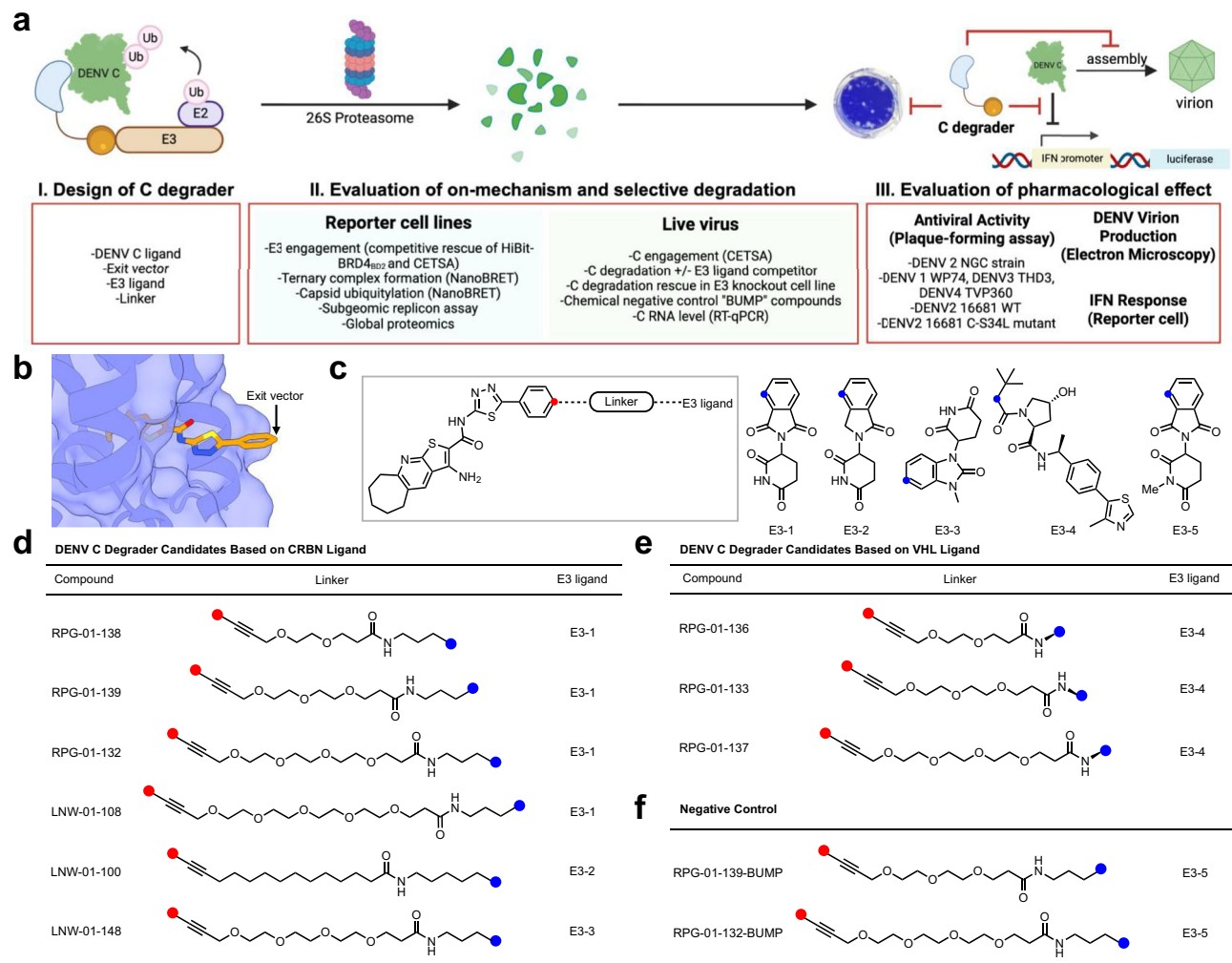

**Fig. 1 | Design and structures of DENV C degraders. a** Outline for development of DENV C degraders. Created in BioRender. Chakravarty, A. (2026) https://BioRender. com/jtuj7ui. **b** The *para* position of the phenyl ring in ST148 was chosen as the exit vector based on the orientation of ST148 in the DENV C dimer (PDB:6VG5). **c** Design of ST148-based bivalent small-molecule degraders and selected ligands of E3

ligases. E1-1, -2, and -3 engage CRL4$^{CRBN}$; E3 and -4 engage CRL2$^{VHL}$; E3-5 has the CRL4$^{CRBN}$ ligand modified to lose interaction with CRBN. Structures of degrader candidates were synthesized and grouped here by **d** CRBN ligands, **e** VHL ligand, and **f** negative control with reduced CRBN interaction.

deficient for CRBN or when the CRBN-binding moiety is modified to reduce interaction with CRL4$^{CRBN}$. Third, we examined whether C is ubiquitinated in the presence of RPG-01-139 and RPG-01-132. To monitor intracellular engagement of CRL4$^{CRBN}$ by RPG-01-139 and RPG-01-132, we used a previously reported competitive displacement assay in which engagement of CRBN results in protection of a HiBiT-BRD4$_{BD2}$-fusion protein[43] from CRL4$^{CRBN}$-mediated degradation in the presence of dBET6, a pan-BET bromodomain degrader[5] (Supplementary Fig. 3). RPG-01-132 showed concentration-dependent engagement of CRBN as evidenced by increased luminescence produced by the HiBiT-BRD4$_{BD2}$-fusion protein (Supplementary Fig. 3d). RPG-01-139 exhibited a more modest ability to compete with dBET6 for binding to CRL4$^{CRBN}$ (Supplementary Fig. 3b), consistent with its lesser effect on C abundance in the initial screen (Fig. 2b). We also conducted cellular thermal shift assays (CETSA) to confirm the engagement of both CRBN and C protein by RPG-01-139 and RPG-01-132. Thermal stabilization of CRBN was observed in uninfected and DENV2-infected Huh7.5 cells at ~51 kDa in cells treated for 1 h with 10 μM of lenalidomide, RPG-01-139, or RPG-01-132 at 51.9 °C, and 55.2/55.8 °C (Supplementary Fig. 4a, c). The DMSO, ST148, RPG-01-139-BUMP, and RPG-01-132-BUMP samples exhibited no evidence of CRBN engagement, as expected (Supplementary Fig. 4a). To examine the engagement of intracellular C in the

context of viral infection, we infected Huh7.5 cells with DENV2 at an MOI of 1 for 1 h and allowed viral replication to continue for 24 h to accumulate sufficient target protein for detection. We then performed CETSA on the infected cells using the same conditions used to assess CRL4$^{CRBN}$-engagement. As expected, DMSO and lenalidomide did not influence the thermal stabilization of C; however, parental inhibitor ST148, RPG-01-139, and RPG-01-132, and the negative control compounds, exhibited stabilization of C at 50.1 °C (Supplementary Fig. 4b, c). These results confirm that RPG-01-139 and RPG-01-132 engage C intracellularly.

To establish that the CRL4$^{CRBN}$ engagement detected in the competitive displacement and CETSA assays is required for the degrader activities of RPG-01-139 and RPG-01-132, we compared their effects on intracellular C protein in wild-type Huh7.5 cells and Huh7.5 cells that had been genetically modified to lack CRBN (Huh7.5-CRBN$^{-/-}$; Supplementary Fig. 5). Huh7.5 and Huh7.5-CRBN$^{-/-}$ cells were infected (DENV2 NGC, MOI 0.5) and then treated with a range of concentrations of RPG-01-139 and RPG-01-132 for 24 h, after which C in cell lysates was analyzed by Western blot (Fig. 3a, b). RPG-01-139 (DC$_{50}$ 3.4 μM and $D_{max}$ 64%, Fig. 3c) and RPG-01-132 (DC$_{50}$ 2.4 μM and $D_{max}$ 84%, Fig. 3d) reduce DENV2 C in a concentration-dependent manner. This activity is largely lost in Huh7.5-CRBN$^{-/-}$ cells (Fig. 3b) and is not observed for

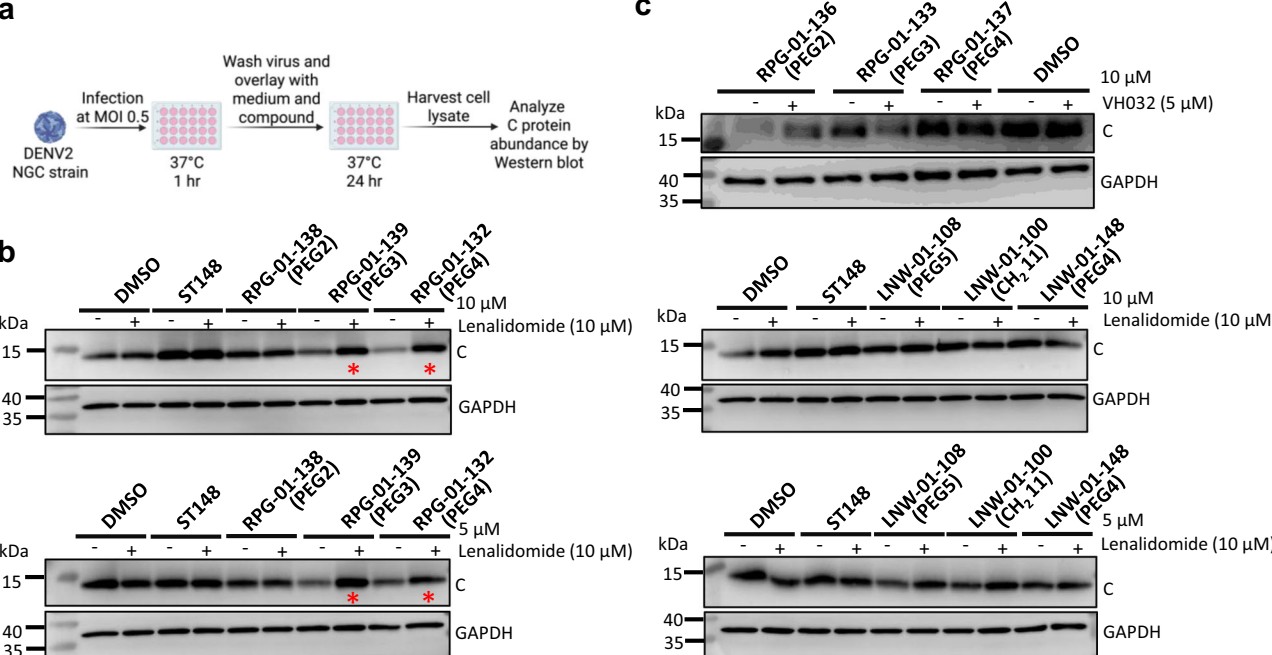

**Fig. 2 | Initial screening of ST148-based DENV degraders. a** Schematic representation of the assay utilized to evaluate the effect of candidate DENV C degraders in virus-infected cells. Created in BioRender. Chakravarty, A. (2025) https://BioRender.com/kcmays9. **b, c** Western blot analysis of intracellular C in DENV2 New Guinea C (NGC)-infected cells treated with the degrader candidates with and without competing E3 ligase ligand as indicated. **b** CRL4$^{CRBN}$-based degraders screened at 10 μM (upper panel) and 5 μM (lower panel) with 10 μM lenalidomide as the competitor. Asterisks indicate candidates showing lenalidomide-sensitive activity. **c** CRL2$^{VHL}$-based degraders screened at 10 μM with 5 μM VH032 as the competitor. We did not test concentrations above 5 μM because VH032 alone has modest antiviral activity at 10 μM, which prevents full rescue of viral replication in the presence of the VHL-based degraders. Representative results are shown from $n = 3$ independent experiments. Densitometric quantification of the blots in Fig. 2 is provided in Supplementary Fig. 2a–e. Source data are provided as a Source Data file.

parental inhibitor ST148 (Supplementary Fig. 6a). The dependence of degrader activity on CRL4$^{CRBN}$ was further demonstrated with negative control compounds, RPG-01-139-BUMP and RPG-01-132-BUMP These two negative control compounds are methylated on the glutarimide moiety to reduce CRBN-binding, which we confirmed in the HiBiT-BRD4$_{BD2}$ competitive displacement (Supplementary Fig. 3c, e) and CETSA assays (Supplementary Fig. 4a, c) but still engage C comparably to the respective C degraders and ST148 in the CETSA assay (Supplementary Fig. 4b, c). RPG-01-139-BUMP and RPG-01-132-BUMP had no effect on C abundance in DENV2-infected cells (Fig. 3b and Supplementary Fig. 7), demonstrating that the effects of RPG-01-139 and RPG-01-132 require CRBN-engagement. Further supporting a TPD mechanism, we found that the presence of MLN4924, a neddylation inhibitor that blocks NEDD8-dependent activation of CRL4$^{CRBN}$, prevented degradation of C by RPG-01-139 and RPG-01-132 (Fig. 3e, f).

As additional confirmation of the TPD mechanism, we conducted assays to detect ternary complex formation and ubiquitination of C and to monitor the effects of RPG-01-132 in these assays. We utilized a commercial NanoBRET assay to monitor ternary complex formation by CRBN, RPG-01-132, and an ectopically expressed C-nLuc fusion protein in HEK293 cells (Supplementary Fig. 8a). Cells co-transfected with C-nLuc and HaloTag-CRBN fusion protein (HT-CRBN) expression plasmids were treated with either DMSO or RPG-01-132 along with the NanoBRET 618 ligand for 4 h, at which time a NanoBRET Nano-Glo substrate was added, and both donor (nLuc: 460 nm) and acceptor (HaloTag: 618 nm) signals were measured. Consistent with ternary complex formation, we observed concentration-dependent BRET signal in the presence of RPG-01-132 (Supplementary Fig. 8b–d). These ternary complexes appear competent for TPD, as evidenced by rescue of the C-nLuc luminescence with the inclusion of proteasome inhibitor MG-132 (Supplementary Fig. 8e–g). This was corroborated by NanoBRET-based detection of ubiquitinated C-nLuc in the presence of

RPG-01-132, but not RPG-01-132-BUMP (Supplementary Figs. 8a and 9). Densitometric quantification of the blots in Fig. 3b is provided in Supplementary Fig. 7. These experiments demonstrate that RPG-01-132 forms a productive ternary complex with CRBN and DENV2 C and causes concentration-dependent depletion of DENV2 C via a mechanism that requires engagement of both CRBN and C, neddylation of CRL4$^{CRBN}$, and proteasome activity. Taken together, they validate DENV C as a target for TPD and establish assays to enable the development of DENV C degraders.

## DENV C degraders exert antiviral activity through targeted degradation of C

To assess the antiviral activity of RPG-01-132 and RPG-01-139, we harvested culture supernatants from the experiments depicted in Fig. 3a and conducted viral plaque formation assays to quantify the yield of infectious progeny virus that had been released to the culture supernatant. Both RPG-01-132 (EC$_{50}$ of 0.47 ± 0.1 μM) and RPG-01-139 (EC$_{50}$ of 0.54 ± 0.02 μM) exerted potent, concentration-dependent antiviral activity (Fig. 4a) in the absence of cytotoxicity (Supplementary Fig. 10). Antiviral activity for RPG-01-132 and RPG-01-139 was well-correlated with the C degradation activity observed in Fig. 3, being absent in Huh7.5-CRBN$^{-/-}$ cells (Fig. 4a) and partially reversed when the neddylation inhibitor MLN4924 was added alongside the degrader (Fig. 4b). RPG-01-132-BUMP and RPG-01-139-BUMP, which can bind to C but not to CRBN, showed very limited (if any) antiviral activity compared to the respective degraders (Fig. 4a) or compared to ST148 (Supplementary Fig. 6b). These observations are all consistent with the idea that the antiviral activity of these degraders is due to targeted degradation of C.

Since the degrader antiviral activity is notably observed at concentrations that cause less than 50% depletion of intracellular C, this could suggest that even a modest depletion of C is sufficient to block the assembly of DENV virions. Alternatively, this might indicate that

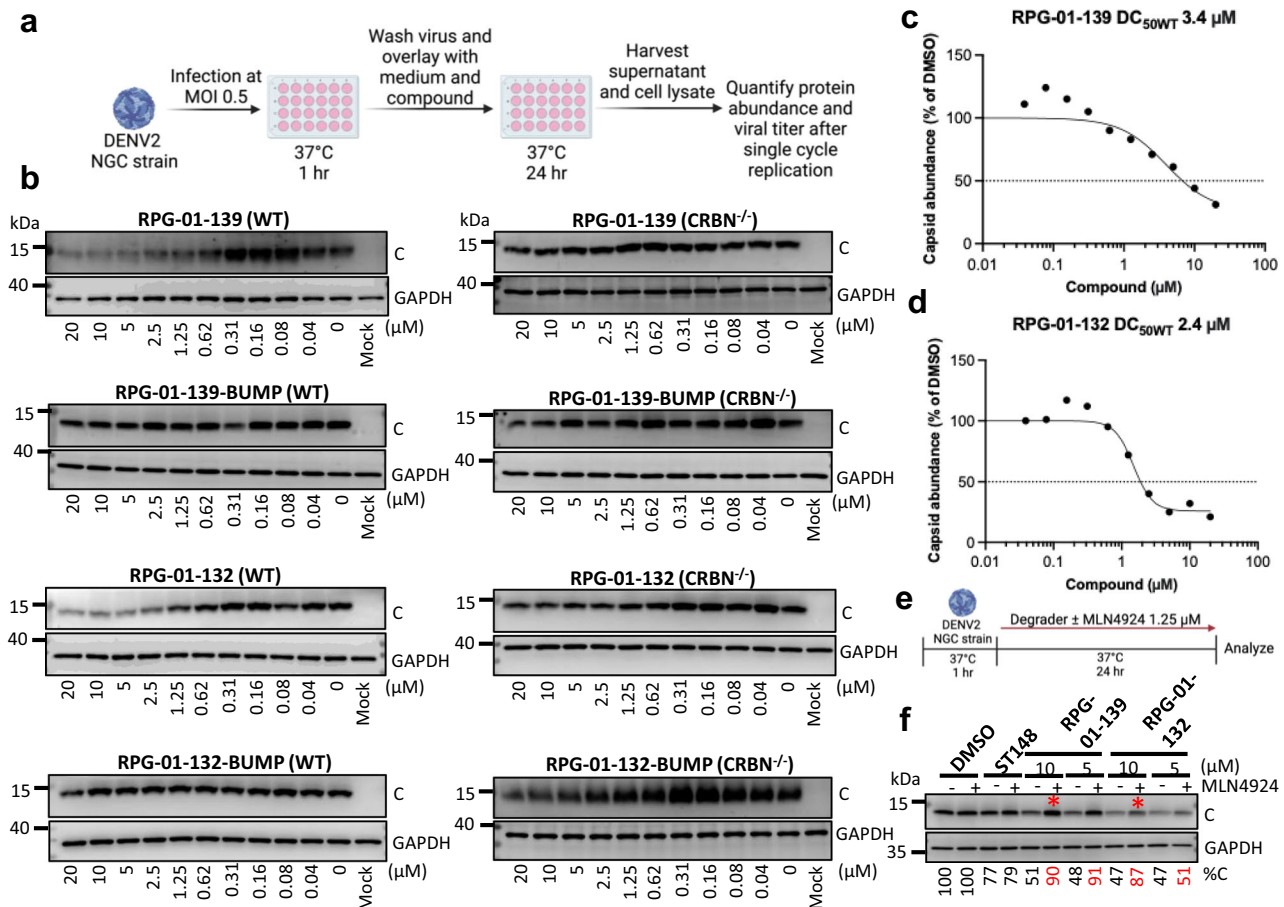

**Fig. 3 | Intracellular depletion of DENV2 C by the C degrader candidates is CRBN- and neddylation-dependent. a** Schematic of assay to measure $DC_{50}$, $D_{max}$, and antiviral $EC_{50}$. Cells were infected with DENV2 NGC at an MOI of 0.5 and treated with RPG-01-139 or RPG-01-132, or their respective negative controls, at the indicated concentrations for hours 1–24 post-infection. Cells were lysed for Western blot analysis, and culture supernatants were collected for quantification of infectious progeny virus by viral plaque formation assay. Created in BioRender. Chakravarty, A. (2025) https://BioRender.com/y0k4zgo. **b** Western blots show concentration-dependent depletion of C in the presence of RPG-01-139 and RPG-01-132. This activity is not observed with negative control compounds RPG-01-139-BUMP and RPG-01-132-BUMP or when the experiment is conducted in Huh7.5-CRBN$^{-/-}$ cells. Representative data are shown for $n = 2$ independent experiments.

Densitometric quantification of the blots is provided in Supplementary Fig. 7. Nonlinear regression analysis was performed using data combined from the two independent experiments to determine the $DC_{50}$ value, defined as the concentration at which capsid abundance is at 50% of that observed for the DMSO control for **c** RPG-01-139 (WT) and **d** RPG-01-132 (WT). **e** Schematic of experiment examining the dependence of C depletion on neddylation. Cells were infected as in (**a**), but the neddylation inhibitor MLN4924 was added along with the degrader for hours 1–24 post-infection. Created in BioRender. Chakravarty, A. (2026) https://BioRender.com/htu70tl. **f** Depletion of C in the presence of RPG-01-139 and RPG-01-132 is partially lost in the presence of MLN4924, indicating that activation of CRL4$^{CRBN}$ by neddylation is required for C degrader activity. The representative data are shown from $n = 2$ independent experiments. Source data are provided as a Source Data file.

RPG-01-132 and RPG-01-139 exert antiviral activity through mechanisms other than loss of capsid function. For example, any mechanism inhibiting replication of the viral RNA genome would also be expected to cause depletion of all viral proteins, including C, over time since the template for translation of these proteins would be reduced. To test our hypothesis that RPG-01-132's antiviral activity is due to specific loss of C, we conducted additional experiments to rule out more general mechanisms. First, we reasoned that if RPG-01-132's antiviral activity is due to a more general effect on replication, this would be reflected in depletion of other DENV proteins and not only C. Experiments examining the effect of RPG-01-132 on the abundance of envelope (E), NS4B, and NS5 detected no changes (Supplementary Fig. 11), confirming that the depletion is specific for C and disfavoring the idea that RPG-01-132 has a general effect on viral replication or viral RNA. As another, more explicit test of whether RPG-01-132's antiviral activity is due to more general effects on viral replication, we examined whether RPG-01-132 changes the steady-state abundance of the viral genomic RNA and found no effect on RNA abundance (Supplementary Fig. 12). We also corroborated that RPG-01-132 does not affect translation or replication

of the viral RNA using a DENV2 subgenomic replicon system[44], which lacks the viral structural genes, including C, and is a well-established model for studying DENV RNA translation and replication in the absence of the steps of viral entry and virion assembly. Whereas the NS4B inhibitor NITD688 caused a reduction in replicon activity, RPG-01-132 and ST148 had no effect in this model (Supplementary Fig. 13), consistent with the idea that their antiviral activities are capsid-specific. Together, these experiments rule out the possibility that RPG-01-132's antiviral activity is due to more general effects on DENV replication and support the hypothesis that its effect on DENV is due to specific loss of C protein.

## GSPT1 degradation does not contribute to the antiviral activity of RPG-01-132

Thalidomide and its derivatives are molecular glue degraders that cause CRL4$^{CRBN}$-mediated polyubiquitination and subsequent proteasomal degradation of multiple, well-established neosubstrates, including the Ikaros family zinc-finger proteins 1 and 3 (IKZF1/3)[45] and the translation termination factor G1 to S phase transition 1 (GSPT1)[46].

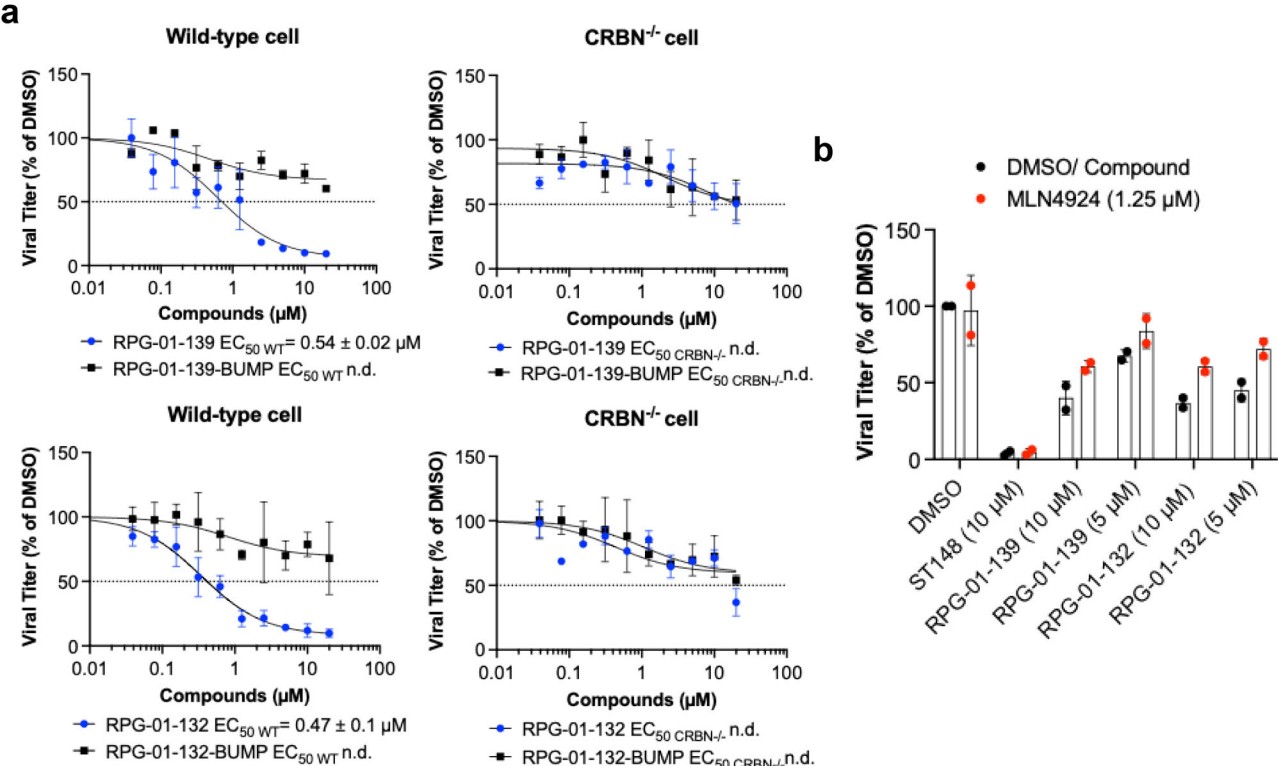

**Fig. 4 | Antiviral activity of degrader candidates depends on the targeted degradation of the capsid. a** Viral plaque assays quantifying infectious progeny virus in cell culture supernatants corresponding to the experiments depicted in Fig. 3a. Antiviral $EC_{50}$ values were determined by nonlinear regression analysis using data from three independent biological experiments ($n = 3$) for RPG-01-139 and RPG-01-132 in wild-type cells. All other conditions were assessed in two independent biological experiments ($n = 2$) and are shown for comparison. Data are presented as mean viral titers normalized to DMSO; error bars represent standard deviation. **b** Antiviral activity of RPG-01-139 and RPG-01-132 in the presence or absence of the neddylation inhibitor MLN4924 (1.25 μM). Data shown are from two independent biological experiments and are presented as means ± standard deviation, normalized to DMSO. No statistical analyses were performed due to the limited number of biological replicates ($n = 2$). Source data are provided as a Source Data file.

Heterobifunctional degraders utilizing thalidomide-based CRBN-targeting ligands may retain molecular glue degrader activity against these proteins and/or acquire additional, unanticipated neosubstrates whose depletion contributes to antiviral activity. We therefore conducted mass-spectrometry-based proteomics experiments to profile the substrate repertoires of RPG-01-132 and RPG-01-139 in an unbiased fashion. Following treatment of MOLT4 cells with 5 μM RPG-01-132 or RPG-01-139 for 8 h, we observed that both RPG-01-132 and RPG-01-139 deplete IKZF3 and GSPT1 (Supplementary Fig. 14a, b). RPG-01-132 exhibits better selectivity with fewer targets than RPG-01-139, with GSPT1 as its main "off-target." Since GSPT1 inhibition is known to be antiviral for several DNA and RNA viruses[47–49], we examined if GSPT1 degradation contributes to the antiviral activity exerted by RPG-01-132 by comparing its GSPT1 $DC_{50}$ and antiviral $EC_{50}$ values with those of CC-90009[50] and SJ6986[51], two potent and selective GSPT1 degraders. CC-90009 and SJ6986 exhibited $DC_{50}$ against GSPT1 in the nanomolar range in Huh7.5 cells (Supplementary Fig. 14c). In contrast, RPG-01-132 is a much weaker degrader of GSPT1, with a $DC_{50}$ value of 7.2 μM (Supplementary Fig. 14d, e). At this concentration, RPG-01-132 reduces viral titer by ~80% (Fig. 4). In contrast, at a concentration of 40 nM CC-90009, corresponding to >10-fold more than its GSPT1 $DC_{50}$ ($DC_{50}$ 3.7 nM), the reduction in viral yield is less than 20% (Supplementary Fig. 14c, g). Based on this, we conclude that although depletion of GSPT1 can have a very modest antiviral effect, it is not a major source of RPG-01-132's antiviral activity since we would otherwise expect to see comparable (if not much higher) levels of antiviral activity for CC-90009 at 10x its $DC_{50}$ concentration.

## DENV C degrader RPG-01-132 has differentiated pharmacology compared to ST148 and affects both assembly and non-assembly functions

We next sought to examine how RPG-01-132 affects DENV replication and C function. Flaviviruses, including DENV, assemble by budding into the endoplasmic reticulum (ER) lumen. Transmission electron microscopy of negative-stained, thin-sectioned cells enables visualization of the overall size, shape, and localization of DENV particles, which appear as roughly spherical, ~50 nm structures within the infected cell. To visualize the effects of pharmacologically induced degradation of C on DENV particle production, we infected Huh7.5 cells for 1 h with DENV2 at an MOI of 20 to maximize the number of infected cells. Following this initial 1 h infection period, the cells were washed to remove excess inoculum and overlaid with medium containing ST148 or RPG-01-132 for 24 h. The cells were then fixed, and negative-stained thin sections were prepared and imaged by transmission electron microscopy (Fig. 5a). Mock-treated cells had no virions as expected, whereas DENV-infected cells exhibited viral particles in both endoplasmic reticulum stacks and Golgi vesicles. ST148 has previously been shown to cause accumulation of DENV2 particles in ER stacks and the release of fewer infectious virions to the culture supernatant due to effects on virion assembly and/or release[52]. Consistent with this prior report, we qualitatively observed that ST148-treated, DENV2-infected cells had large ER stacks of diameter up to 1000 nm filled with clusters of viral particles and relatively fewer viral particles that had progressed to Golgi vesicles when compared to the DENV2-infected controls treated with DMSO (Fig. 5a). In contrast, the

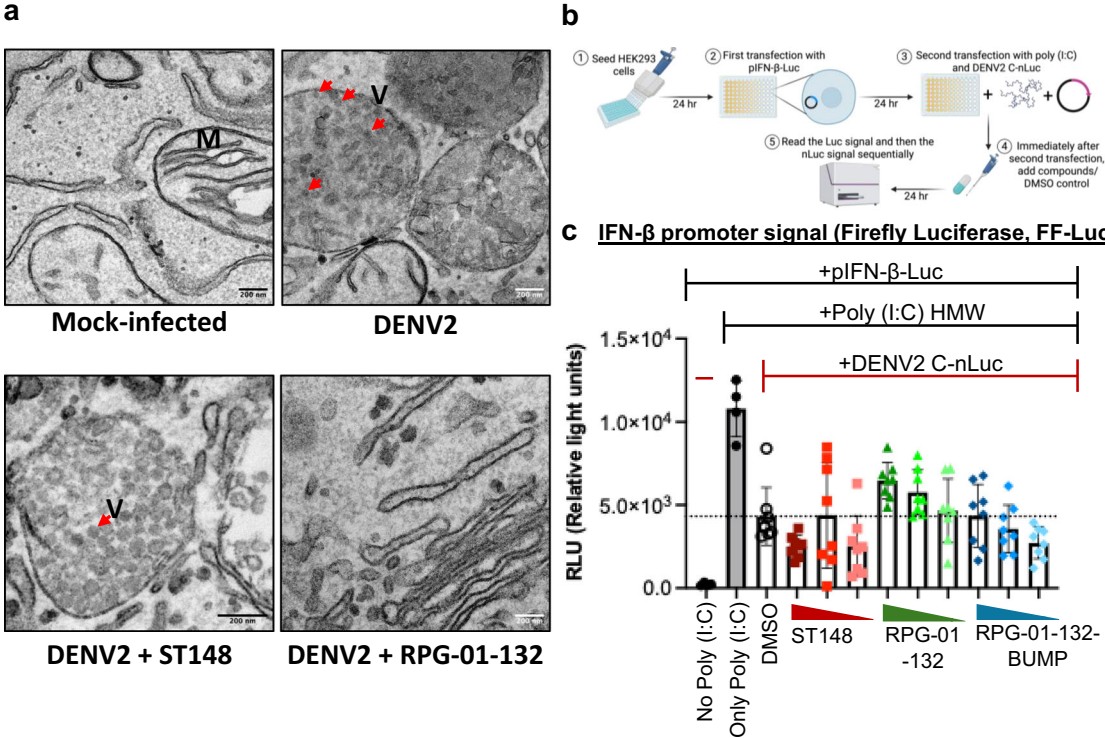

**Fig. 5 | RPG-01-132 inhibits multiple functions of the dengue capsid protein. a** Negative-stained thin sections were imaged by transmission electron microscopy to visualize the effects of inhibitor vs. degrader treatment on virion assembly. Clockwise from upper left, samples are mock-infected with DMSO vehicle, DENV2 NGC-infected with DMSO vehicle, DENV2 NGC-infected treated with RPG-01-132, and DENV2 NGC-infected treated with ST148. Red arrows indicate the presence of DENV2 virions. M = mitochondria, V = virions. For each condition, two grids were screened, and three representative images are shown here and in Supplementary Fig. 15. The experiment was performed twice independently with similar results. The scale bar is 200 nm. **b** Schematic of the experimental design to investigate RPG-01-132's effect on the antagonism of the type I interferon response. In this dual luciferase system, a firefly luciferase reporter gene (FF-Luc) is regulated by the IFN-β promoter, and C is ectopically expressed as a fusion protein with nanoluciferase (C-nLuc) to allow monitoring of C abundance. ST148, RPG-01-132, and RPG-01-132-BUMP were used at 0.156, 0.312, or 0.625 μM. Created in BioRender. Chakravarty, A. (2026) https://BioRender.com/ownzyzs. **c** Only RPG-01-132 rescues IFN-β promoter activity in a concentration-dependent manner; ST148 and RPG-01-132-BUMP do not, indicating that this effect is dependent on targeted degradation of DENV C. The IFN-β reporter assay measures luminescence output as relative light units (RLU). Data shown are from one representative experiment, with technical replicates plotted as individual data points. Bars represent mean values, and error bars indicate standard deviation. The experiment was repeated three times independently with similar results, which are shown in Supplementary Fig. 17b, c. Data for nLuc-based detection of C−nLuc abundance are provided in Supplementary Fig. 17d−f. Source data are provided as a Source Data file.

presence of RPG-01-132 results in the detection of very few viral particles in the ER or Golgi and the appearance of vesicular structures resembling autophagosomes and/or lysosomes (Fig. 5a). This suggests a block in viral assembly caused by the limited availability of C protein and is a phenotype that differs strikingly from that induced by ST148. Thus, ST148 and RPG-01-132 appear to have different mechanisms despite using the same chemical moiety to interact with C. Additional samples in which DENV2-infected cells were treated with RPG-01-132-BUMP or CC-90009 most closely resemble the DENV2-infected, DMSO-treated sample, with viral particles present in Golgi vesicles and ER stacks (Supplementary Fig. 16). This supports our interpretation that the effects of RPG-01-132 are due to targeted degradation of DENV C and not due to targeted degradation of GSPT1.

We additionally sought to examine if RPG-01-132 can affect other functions of C that are separate from its role as a structural component of virions. In a recent study of a non-assembly role of flaviviral capsid proteins, Hobman and colleagues[23] showed that capsid proteins from diverse mosquito-borne flaviviruses inhibit induction of the IFN-β response due to their interaction with TRIM25, which suppresses RIG-I polyubiquitination and activation. We examined whether RPG-01-132 affects this immune-evasive function of C following the experimental design outlined in Fig. 5b, using high molecular weight poly(I:C) to activate expression of a firefly luciferase (FF-Luc) reporter under the control of the IFN-β promoter. Since NS4B, NS5, and other DENV

proteins have also been shown to antagonize interferon responses[53–55], we ectopically expressed a DENV2 NGC C-nLuc fusion protein instead of infecting with live virus. This system allowed us to monitor C abundance and degrader activity by monitoring luminescence of the nLuc reporter while also avoiding confounding results due to multiple, overlapping mechanisms of IFN antagonism. We first transiently expressed the C-nLuc fusion protein in HEK293 cells and titrated RPG-01-132 (10 nM–10 μM) to determine the concentration range at which C degradation occurs in this system. While we observed stabilization of C-nLuc at higher concentrations of the degrader, perhaps consistent with a hook effect, we observed consistent depletion of approximately 50% of C-nLuc in the concentration range between 0.625-0.156 μM (Supplementary Fig. 17a). We also verified that this concentration range of RPG-01-132 causes depletion of untagged DENV2 capsid ectopically expressed in HEK293 (Supplementary Fig. 17g), demonstrating that nLuc activity reflects C-nLuc abundance in this system.

As outlined in Fig. 5b, HEK293 cells were first transfected with the pIFN-β-Luc reporter plasmid and 24 h later, transfected with high molecular weight poly(I:C) and the C-nLuc plasmid. ST148, RPG-01-132, RPG-01-132-BUMP were also added at this time at the indicated concentrations with DMSO as a negative control. Consistent with the published report[23], we observed that ectopic expression of C caused a reduction in IFN promoter activity 24 h later. IFN-β promoter activity was rescued in the presence of RPG-01-132 (Fig. 5c and Supplementary

Fig. 17b, c), suggesting that C's antagonism of the IFN response can be blocked by targeted degradation of C. We attempted to verify that C-nLuc had been depleted by RPG-01-132 but were unable to detect differences in nLuc luminescence in the dual firefly luciferase-nLuc assay (Supplementary Fig. 17d–f). While this may seem to indicate limited degradation of C in this experiment, this may be explained by RPG-01-132's partial degradation profile[56] ($D_{max}$ 84% in Huh7.5, Fig. 3d and Supplementary Fig. 17g) and a strong contribution of non-degradable C to the nLuc signal in this system. In addition, the dual FF-Luc-nLuc assay may have a reduced ability to detect small decreases in nLuc activity since we do observe reduced nLuc activity and reduced C-nLuc protein by Western blot (Supplementary Fig. 17a, g) in the control experiments conducted at these concentrations of RPG-01-132. Consistent with the idea that RPG-01-132's rescue of the IFN response in this system is due to targeted degradation of C, neither ST148 nor RPG-01-132-BUMP restored activation of the IFN promoter in the presence of C (Fig. 5c and Supplementary Fig. 17b, c). This suggests that degradation of C can block its antagonism of the IFN response, whereas ST148's stabilization of C dimers cannot, at least not in this ectopic expression system.

### C degraders degrade ST148-resistant C variants and show potent antiviral activity

The ability of viruses to generate genetic diversity limits the activity spectrum of most DAAs and enables the rapid evolution of drug resistance. We have previously observed that conversion of antiviral inhibitors targeting the hepatitis C virus (HCV) NS3-4A protease and DENV envelope protein into antiviral degraders resulted in improved resilience to point mutations that reduce drug-binding and broadened activity spectrum[5,6]. We have posited that this is due to conversion from occupancy-driven pharmacology, which is driven solely by target-binding, to event-driven pharmacology, which does not require tight, stoichiometric binding; however, more examples are needed to test the generalizability of this strategy. Having established RPG-01-132 as an antiviral degrader that acts by inducing targeted degradation of C, we next turned to comparing its activity versus ST148 against strains representative of the four dengue serotypes (DENV1-4) and against a point mutant known to confer resistance to ST148.

A published high-resolution co-crystal structure of ST148 bound to C[41] demonstrated that two molecules of ST148 bind at the dimer interface of the stabilized C tetramer in a pocket formed by 7 residues (Thr30, Phe33, Leu35, Met37, Leu38, Leu50, and Phe53), of which 5 are not conserved across DENV1, 3, and 4[41] (Fig. 6a). Variability at these positions in DENV serotypes 1, 3, and 4 is correlated with reduced sensitivity to ST148 compared to DENV2, with DENV4 being the most divergent and least sensitive in prior studies (Fig. 6b)[41]. To examine if the TPD mechanism of RPG-01-132 is associated with a broadened spectrum of activity, we conducted antiviral activity assays comparing ST148 and RPG-01-132 following the experimental scheme outlined in Fig. 3a and using a concentration of 10 μM for both ST148 and RPG-01-132 based on previous characterization of ST148's activity against DENV1, 3, and 4[40,41]. Parallel samples were conducted with and without lenalidomide as a competitor for CRBN-binding to demonstrate on-mechanism activity of the degrader in these experiments (Fig. 6c). RPG-01-132 exhibited less potent activity against DENV2 compared to ST148. This is likely attributable to decreased cell permeability of the degrader compared to the parental inhibitor, which we confirmed experimentally (Supplementary Table 1) and which is consistent with other heterobifunctional degraders compared to their respective parental targeting ligands. Importantly, whereas ST148 exhibits selectivity for DENV2 over DENV1, DENV3, and DENV4, RPG-01-132 exhibits consistent activity across all four serotypes. This is concordant with the idea that event-driven pharmacology can mitigate the need for tight, stoichiometric binding required by antivirals with occupancy-driven pharmacology.

Next, we examined RPG-01-132's efficacy against an ST148-resistant DENV2 mutant. Serial passage of DENV2 in the presence of ST148 resulted in the identification of a single mutation at position 34 from serine to leucine (S34L), which is sufficient to confer resistance[40]. While Ser34 does not directly contact ST148 in the co-crystal structure, its side chain hydroxyl forms hydrogen bonds to the amide backbone at Gly36 and Met37 (Fig. 6c). Loss of these hydrogen bonds was hypothesized to affect the hydrogen bond between the backbone carbonyl of Phe33 with ST148, resulting in reduced ST148-binding and/or reduced formation of the stabilized tetramer. We confirmed that the DENV2 C S34L mutation confers loss of sensitivity to ST148 in DENV2 strain 16881, resulting in a large shift in $EC_{50}$ (Fig. 6d). In contrast, titration of RPG-01-132 against DENV2 16881 wild-type and C-S34L viruses revealed comparable levels of C degradation ($DC_{50}$ wild-type 3 μM and $DC_{50}$ S34L 1.85 μM) and antiviral activity (wild-type, $EC_{50}$ wildtype $1 \pm 0.5$ μM; S34L mutant, $EC_{50}$ $0.58 \pm 0.28$ μM) (Fig. 6e–i). Collectively, our experiments show that conversion of ST148 to a degrader mechanism results in new pharmacology accompanied by a broadened activity spectrum.

## Discussion

The development of TPD-based agents for oncology and inflammation has advanced tremendously, advancing well over thirty compounds to the clinic and establishing a road map for the development of new therapeutic agents with this pharmacological mechanism[57]. Application of this strategy to develop novel antivirals has been less rapid, in part due to distinct challenges rooted in the unique biology of viral infections. These are illustrated in our efforts to apply TPD to DENV C. As building blocks for new virions, DENV C and other viral structural proteins are often highly abundant, and the level of depletion required for antiviral activity may be difficult to predict and also difficult to achieve. Subcellular localization can present an additional challenge because many viral proteins are sequestered within large, macro-molecular assemblies or in virus-induced subcellular structures, including proteinaceous "factories" and membranous "organelles." This may "hide" viral proteins from the ubiquitin-proteasome system and/or sterically interfere with the formation of a productive ternary complex. In part to ensure that our DENV C degrader development effort advanced bivalent compounds that were "on-mechanism," we established multiple assays (Fig. 1a) (1) to monitor engagement by C and the E3 ligase (Supplementary Figs. 3 and 4); (2) to assess E3 ligase dependence (Figs. 2, 3 and Supplementary Fig. 2); (3) to monitor ubiquitination of DENV C (Supplementary Fig. 9); and (4) to detect antiviral activity occurring via a TPD mechanism (Fig. 4). This full panel of tools ultimately enabled us to synthesize and screen a series of heterobivalent molecules, culminating in the identification of two molecules, RPG-01-139 and RPG-01-132, that degrade DENV C in a CRL4[CRBN]-dependent manner. While RPG-01-132 exhibited "on-mechanism" activity across all assays in our panel, RPG-01-139 lacked significant activity in the DENV C ubiquitination assay and limited activity in a competitive assay measuring CRL4[CRBN]-engagement despite exhibiting on-mechanism activity in other assays. This may reflect the challenges of detecting marginal activity at the start of a degrader development project and argues for the use of multiple, complementary assays to screen degrader candidates. We note that our initial collection of degrader candidates included several with a VHL-targeting ligand and that we were able to identify candidates with on-mechanism activity even within this very limited set (Fig. 2c). Linker SAR differed for CRL4[CRBN]- versus CRL2[VHL]-based TPD activities. CRL2[VHL]-based degraders exhibited greater activity with the short PEG2 linker (RPG-01-136), with longer linkers associated with losses in activity (RPG-01-133 for PEG3 and RPG-01-137 for PEG4). In contrast, the PEG3 and PEG4 linkers of RPG-01-132 and RPG-01-139 were optimal for the CRL4[CRBN]-based series, illustrating the need to optimize linker and E3 ligands to function together. Although we did not pursue the VHL series due to its

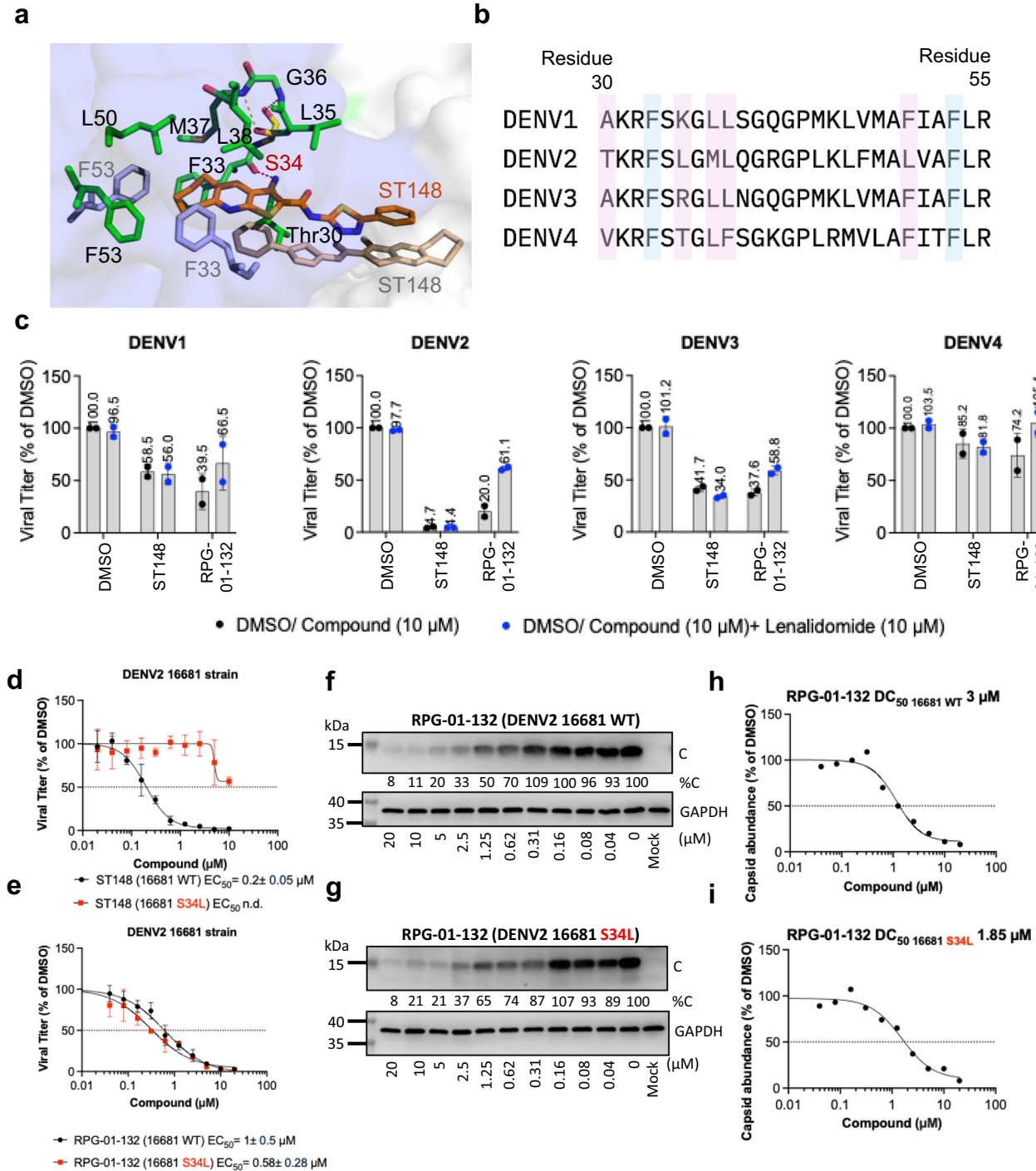

**Fig. 6 | RPG-01-132 has consistent activity against the four DENV serotypes and the ST148-resistant DENV2 C S34L mutant. a** Orientations of two molecules of ST148 with the DENV2 C tetramer (PDB: 6VG5)[41]. The binding pocket of one ST148 molecule (shown in orange) is shown in detail. **b** Alignment of amino acid sequences of DENV1 WP74, DENV2 NGC and 16681, DENV3 THD3, and DENV4 TVP360. Five residues (30, 35, 37, 38, and 50) are variable, and two residues (F33 and F53) are conserved across all four serotypes. **c**–**i** Antiviral activity was measured against DENV1 WP74, DENV2 NGC, DENV3 THD3, DENV4 TVP360, DENV2 16681, and DENV2 16681 C-S34L strains following the experimental outline in Fig. 3a. **c** RPG-01-132 exhibits consistent, CRBN-dependent antiviral activity across all four DENV serotypes. Black and blue symbols represent treatment with DMSO/compound alone or co-treatment with DMSO/compound and lenalidomide, respectively. Each data point represents an independent biological replicate, with two technical replicates averaged per experiment. Bars represent means normalized to DMSO ±

standard deviation from *n* = 2 independent biological replicates. No statistical hypothesis testing was performed for this panel. (d) DENV2 16681 is sensitive to ST148 (EC50 0.2 ± 0.05 μM), but the DENV2 16681 C-S34L mutant is not (EC50 could not be determined). (e) RPG-01-132 exhibits comparable antiviral activity against wildtype and ST148-resistant viruses (EC50 1 and 0.6 μM, respectively). **f**–**i** RPG-01-132 exhibits comparable degradation activity against DENV2 16681 and DENV2 16681 C-S34L viruses with DC50 values of 3 and 1.85 μM, respectively. Viral titers in c-e are presented as means normalized to DMSO ± standard deviation. Antiviral EC50 values were determined by nonlinear regression analysis of the data from *n* = 2 independent experiments. Band intensities in f and g were quantified by ImageJ. DC50 values were determined by nonlinear regression analysis of the mean capsid abundance normalized to DMSO from *n* = 2 independent experiments. Source data are provided as a Source Data file.

lower activity in initial screens, our experiments indicate that DENV C can be targeted with degraders utilizing CRL4$^{CRBN}$ or CRL2$^{VHL}$ E3 ligases. Together, this work validates DENV C as a target for TPD-based activity and establishes a technical foundation for the identification and optimization of C-targeting DAAs that act by TPD mechanisms.

Due to the relatively high abundance of DENV C, the threshold of depletion needed for antiviral activity was a major question at the start of our study. In our experiments, which were conducted in Huh7.5 cells, RPG-01-139 and RPG-01-132 exhibited partial degradation profiles[56] with maximum degradation ($D_{max}$) levels of 64% and 84%, respectively, yet they exhibit sub-micromolar antiviral activity. Since we were able to eliminate degradation of GSPT1 or other off-targets as a source of this antiviral activity, the significantly lower EC$_{50}$ values for both compounds compared to their respective DC$_{50}$ values (RPG-01-139: DC$_{50}$ 3.4 μM, EC$_{50}$ 0.47 ± 0.1 μM; RPG-01-132: DC$_{50}$ 2.4 μM, EC$_{50}$ 0.54 ± 0.02 μM) were notable. C's multiple functions may be differentially affected by C degradation, with some capsid-related pathways more affected by depletion than others, both in terms of the susceptibility to the TPD mechanism due to differences in localization, binding partners, or other reasons but also in terms of the impact of reduced C on the viral pathway – for example, the threshold of depletion needed to impact C's role in assembly may differ from that needed to impact a function in signal transduction. Since our study monitored total intracellular C, additional experiments are needed to understand how the partial degradation phenotypes of RPG-01-132 and RPG-01-139 confer the potent, C-specific antiviral activity observed.

A main driving force for interest in using TPD as an antiviral strategy is the potential for its event-driven pharmacology to expand or extend the spectrum of activity typical of most DAAs. Likewise, the observation that TPD-based agents can be more resilient to mutations that confer inhibitor resistance in cancer biology[58-61] stimulated interest in using this approach to avoid (or delay) drug resistance. Since there are limited studies addressing whether this is anecdotally true for antiviral degraders, we examined this explicitly for RPG-01-132. The DENV2 C-S34L mutant is an ST148-resistant mutant selected following serial passage of DENV2 in the presence of ST148[40]. Its mechanism of resistance has not been fully established. While the high-resolution co-crystal structure of wild-type DENV C bound to ST148 suggests that the S34L mutation may indirectly affect drug-binding, intrinsic fluorescence assays revealed no difference in ST148-binding with recombinant wild-type and C-S34L proteins[40], and comparisons of C-binding in the context of DENV2 wild-type and C-S34L viruses have not been reported. Regardless of the mechanism of resistance, ST148 exhibits a drastic loss in efficacy against the C-S34L mutant, whereas RPG-01-132 has similar degradation and antiviral activities against DENV2 16681 wildtype (DC$_{50}$ 3 μM, EC$_{50}$ 1 ± 0.5 μM) and C-S34L (DC$_{50}$ 1.85 μM, EC$_{50}$ 0.58 ± 0.28 μM) viruses (Fig. 6). We observed a similar trend in comparing ST148 and RPG-01-132 against the other DENV serotypes. Its lower cell penetrance notwithstanding, RPG-01-132 exhibits broader spectrum activity with comparable activity against all four DENV serotypes, whereas ST148 shows pronounced selectivity for DENV2. RPG-01-132's activity against the serotypes and the ST148-resistant mutant supports the idea that the event-driven pharmacology of TPD-based antivirals tolerates amino acid differences that more drastically affect occupancy-based antivirals.

Another driving force in our exploration of TPD against DENV C has been the pursuit of antiviral activities that are not currently achievable with antiviral inhibitors. In comparative negative-stain, transmission electron microscopy experiments, we observed markedly different phenotypes for ST148 and RPG-01-132. Consistent with its published stabilization of C tetramers, which interferes with normal particle budding, ST148 caused the accumulation of large arrays of electron-dense viral particles in ER stacks, which corresponded to a large reduction in extracellular release of infectious virions. In contrast, RPG-01-132 treatment resulted in DENV-infected cells that had very few assembled virions. Although we cannot at this time distinguish a direct effect of RPG-01-132 on viral assembly from other potential mechanisms impacting capsid-dependent processes, the pharmacological effect of RPG-01-132 clearly differs from that of ST148. This complete switch in mechanism is, to our knowledge, unlike other described degraders derived from antiviral inhibitors[8,16,62,63], all of which at least partially retain the activity of their respective parental compounds. In line with this, we have previously observed that negative control ("BUMP") compounds that cannot engage the E3 ligase also revert to the mechanism of the parental inhibitor used as a targeting ligand. We do not currently know if the degraders engage DENV C in a manner incompatible with ST148's mechanism or if degradation of DENV C occurs before it can be incorporated into a new virion. The negative control compounds RPG-01-132-BUMP and RPG-01-139-BUMP are devoid of antiviral activity (Figs. 3 and 4), and the phenotype observed for RPG-01-132-BUMP in the negative stain transmission electron micrograph resembles that of the DENV2-infected, DMSO-treated control and not that of ST148 or RPG-01-132. This may also reflect the reduced cell penetrance of the heterobivalent compounds, such that RPG-01-132-BUMP does not accumulate intracellularly at the threshold concentration needed for ST148's mechanism of action, or such that the linker modification alters the interaction of ST148 with the target. Regardless of the underlying reason, the absence of ST148 activity for the DENV C degraders indicates that their antiviral activity is due to target degradation and is well-differentiated from ST148's activity. Prior characterization of ST148 did not examine its effects on non-assembly functions of DENV C. Our experiments showing RPG-01-132's rescue of poly(I:C)-induced IFN-β promoter activity suggest that this immune-evasive activity of C is blocked. This effect of RPG-01-132 is also TPD-mediated, as it was not observed with ST148 or with RPG-01-132-BUMP. Together, these data clearly demonstrate that conversion of ST148 to a degrader mechanism results in new antiviral pharmacology.

In summary, our study demonstrates the feasibility of applying TPD against the multifunctional DENV capsid protein as an antiviral strategy. Despite the challenges posed by the unique biology of viral infections, including spatial and temporal variability of protein expression, differential target abundance, and complex subcellular localization, we demonstrate that RPG-01-132 effectively degrades the capsid protein through an E3 ligase-dependent mechanism, resulting in disruption of capsid-related pathways required for productive infection, including reduced release of infectious virus and rescue of C-mediated antagonism of an interferon response. Potent antiviral activity was observed despite a partial degradation profile, including against DENV1, DENV3, and DENV4 serotype strains and a DENV2 mutant resistant to ST148. The panel of assays we report provides a platform for further development of antiviral degraders targeting C as potential therapeutics but also potentially as tools for dissecting the functions of specific subcellular pools of C. Demonstration of antiviral efficacy in vivo is a necessary next step in the preclinical development of TPD-based antivirals. This will require improvements in potency, cell permeability, and other ADME-PK parameters[64] with iterative rounds of medicinal chemistry to optimize linkers and E3 ligands, and may additionally require the identification of improved ligands for the DENV C protein. Due to species differences in E3 ligases, an animal model capable of evaluating DENV degraders optimized against human E3 ligases and accepted by the DENV field will also have to be established. While success in these endeavors cannot be assumed, optimization of heterobifunctional degraders targeting BRD4, BCL6, estrogen receptor, androgen receptor, and many other targets to clinical stage candidates demonstrates that this preclinical and clinical optimization is possible. This study validates DENV C as a target for TPD DAA development and provides tools needed for this goal.

## Methods

### Cell culture

The mammalian cell lines used in this study were maintained in a 37 °C incubator with 5% $CO_2$. Huh7.5 cells, a gift from Charles Rice (Rockefeller University), and HEK293 cells, obtained from Trung Pham (Stanford University), were maintained in Dulbecco's Modified Eagle's medium (DMEM, Corning #10-013-CV) supplemented with nonessential amino acids, 10 mM HEPES, and 10% fetal bovine serum (FBS). Baby Hamster Kidney cells (BHK), obtained from Eva Harris (University of California, Berkeley), were maintained in Minimum Essential Medium-Alpha (MEM-α, Corning #10-022-CV) supplemented with 10 mM HEPES and 5% FBS. Vero cells were purchased from ATCC and were maintained in DMEM supplemented with nonessential amino acids, 10 mM HEPES, and 10% FBS. C6/36 cells, a mosquito cell line derived from the embryonic tissue of *Aedes albopictus* (Diptera: Culicidae), were purchased from ATCC and maintained in Leibovitz medium (L-15) containing 10% FBS at 28 °C. All cell lines were routinely tested for mycoplasma contamination using a PCR-based assay and were confirmed to be mycoplasma-negative at the time of experimentation.

### Viruses

Dengue virus serotype 1 WestPac 74 (DENV1 WP74), serotype 3 THD3 (DENV3 THD3), and serotype 4 TVP360 (DENV4 TVP360) strains were propagated from virus stocks obtained from Aravinda De Silva (University of North Carolina, Chapel Hill). Dengue virus serotype 2 New Guinea C (NGC) was obtained from Lee Gehrke (Massachusetts Institute of Technology). Dengue virus serotype 2 16681 wild type and S34L mutant virus were obtained from Karla Kirkegaard (Stanford University). DENV1 and DENV2 were propagated at an MOI of 0.1 in Vero and C6/36 cells, respectively. Briefly, the supernatant was collected 4 days post-inoculation upon detection of cytopathic effects. Cell debris was removed by centrifugation, and the virus stocks were aliquoted and stored at −80 °C until further use. BHK21 cells cultured in MEM-α with 5% FBS were used to determine titers of DENV1 WP74, DENV2 NGC, DENV2 16681, and DENV4 TVP360. Vero cells cultured in DMEM with 2% FBS were used to titer DENV3 THD3. All work with the dengue virus was performed in a biosafety level 2 laboratory (BSL2) with BSL2 enhanced procedures and was reviewed and approved by the Stanford Administrative Panel on Biosafety.

### Generation of CRBN knockout cells

**Plasmids construction.** To generate plasmids for CRBN knockout, the guide RNA sequences were designed using Benchling, and oligos were synthesized from Integrated DNA Technology (IDT). The oligos 5′-caccgCAGGACGCTGCGCACAACAT-3′ and 5′-aaacATGTTGTGCGCAGCGTCCTGc-3′ were cloned into pLenti-CRISPRV2 (Addgene, Plasmid #52961). Using Gibson assembly, the puromycin-resistant gene was replaced with the zeocin-resistant gene to generate the pLentiCRISPRV2-CRBNsg-Zeocin plasmid.

**Lentivirus packaging.** Lentivirus transduction was conducted on designated cell lines to generate knockout cell lines. To package lentivirus stock, the plasmids of interest carrying the transgene were co-transfected with the packaging plasmids ΔVPR, VSV-G, and pAdVAntage into HEK293FT cells using TransIT-LT1 transfection reagent (Mirus Bio, # MIR2304). At 48 h post-transfection, the lentivirus-containing medium was harvested, filtered through 0.45-micron filters, and stored at −80 °C for later transduction.

To generate CRBN knockout Huh7.5 cell lines, the cells were seeded at a density of $2 \times 10^5$ cells/well in 6-well plates one day before transduction and transduced with lentivirus packaged with pLentiCRISPRV2-CRBN-zeocin. 48 h post-transduction, the cells were selected with zeocin at 3 mg/mL for Huh7.5 cells. The decrease in CRBN expression in the pool knockout cells was further confirmed with the Western blot using anti-CRBN antibody. We validated functional knockout of CRBN by showing that dFKBP-1[65], a validated $CRL4^{CRBN}$-dependent degrader of FKBP12, has concentration-dependent degrader activity against FKBP12 in wild-type Huh7.5 cells but not in Huh7.5-CRBN−/− cells (Supplementary Fig. 5).

### Western blotting and antibodies

Whole-cell lysates were prepared by cell lysis in 1X radioimmunoprecipitation assay buffer (Thermo Scientific Chemicals, #J62524AE) containing 1X protease inhibitor cocktail (Apexbio Technology LLC, #K10191). Equal amounts of proteins were separated on an 8-16% Mini-PROTEAN TGX precast gel (Bio-Rad Laboratories, #4561106) and transferred to a polyvinylidene difluoride membrane using Trans-Blot Turbo RTA PVDF Transfer Kit (Bio-Rad Laboratories, #1704272) and a Trans-Blot Turbo Transfer System (Bio-Rad Laboratories, #17001917) according to the manufacturer's protocol. The membrane was blocked for 1 h at room temperature in TBS-Tween 20 (TBS-T) with 3% (wt/vol) bovine serum albumin (BSA, Fisher Bioreagents, #BP9703100), then incubated overnight at 4 °C with a desired dilution of antibodies. After that, the membrane was washed three times in TBS-T, followed by incubation for 1 h at room temperature with HRP-conjugated secondary antibodies. After three washes in TBS-T, the membrane was developed with Pierce SuperSignal West Pico Plus Chemiluminescent Substrate (Thermo Scientific, #34580), and the signal was captured using the ImageQuant 800 CCD Imager (Cytiva). Band intensities were quantified using ImageJ software (https://imagej.net/ij/).

Monoclonal antibody targeting dengue virus type 2 capsid protein (6F3.1 hybridoma) was a gift from John Aaskov (Queensland University of Technology, Brisbane, Australia) or GT574, which was purchased from Thermo Fisher Scientific (#MA5-35908), and used at a ratio of 1:1000 for Western blotting of DENV2 NGC and 16681 C. Mouse monoclonal antibody anti-GAPDH (6C5) was purchased from GeneTex (#GTX28245) and used at a ratio of 1:10,000. Rabbit polyclonal Anti-eRF3/GSPT antibody was purchased from Abcam (#ab126090) and used at a ratio of 1:1000. Rabbit monoclonal antibody anti-CRBN (D8H3S) was purchased from Cell Signaling Technology (#71810) and used at a ratio of 1:1000. Monoclonal antibody targeting dengue virus type 2 envelope protein (4G2 hybridoma) was a gift from Aaron Schmidt (Harvard Medical School) and used at a ratio of 1:1000. Rabbit polyclonal antibody anti-dengue virus NS4B was purchased from GeneTex (#GTX124250) and used at a ratio of 1:3000. Mouse monoclonal antibody anti-dengue virus NS5 (GT353) was purchased from GeneTex (#GTX629446) and used at a ratio of 1:1000. Horseradish peroxidase (HRP)-conjugated goat anti-mouse IgG (#1706516) and goat anti-rabbit IgG (#1706515) were purchased from Bio-Rad Laboratories and used at a ratio of 1:3000.

### NanoBRET assay for degrader-induced CRBN−C ternary complex formation

The coding sequence of DENV2 NGC C was amplified and inserted into the pTwist CMV Puro vector to express NanoLuc at the C-terminus of C protein (Twist Biosciences). HEK293 cells were plated in 6-well plates at $8 \times 10^5$ cells/well and allowed to adhere for 4–6 h at 37 °C, 5% $CO_2$. The cells were then co-transfected at a 1:10 or 1:100 ratio of the pTwist-DENV2C-nLuc plasmid (expressing C-nLuc) with the HaloTag-CRBN plasmid (expressing HT-CRBN, Promega). The transfection was performed using XtremeGeneHP DNA transfection reagent (Roche, #06366244001) according to the manufacturer's protocol. Cells were washed once with 1 mL PBS, trypsinized, and resuspended in Opti-MEM (phenol-red–free) + 4% FBS (Thermo Fisher, #11058021) at $2.2 \times 10^5$ cells/mL. The cell suspension was split into two pools and treated with either DMSO (0.1% final) or HaloTag NanoBRET 618 Ligand (Promega, 100 nM final). 32 μL of cells were dispensed into white 384-well plates (Corning, #3570) and were incubated at 37 °C for 20 h. A

10× compound series in phenol red-free Opti-MEM (e.g., 100 μM for a 10 μM final) and a matched DMSO control were prepared; 4 μL was added to each well and incubated at 37 °C for 4 h. For proteasome-inhibition controls (where indicated), MG-132 (10 μM) was added 30 min before the compound. Immediately before reading, a 5× working solution of NanoBRET Nano-Glo substrate (Promega; prepared as a 1:100 dilution of stock into substrate buffer) was added at 10 μL/well (final volume 50 μL). Plates were read on a CLARIOstar (BMG Labtech), collecting donor emission at 450 nm/80 nm and acceptor emission at 618 nm (610 nm LP). Raw NanoBRET ratios were calculated as acceptor/donor, converted to mBU by ×1000, and the no-ligand values were subtracted to yield the corrected BRET for each condition. A dose-dependent increase in corrected BRET indicates ligand-induced proximity between C-nLuc and HT-CRBN, consistent with ternary complex formation.

### NanoBRET assay for degrader-mediated C ubiquitination

HEK293 cells were plated in 6-well plates with a density of $8 \times 10^5$ cells/well and allowed to adhere and recover for 4-6 h at 37 °C, 5% $CO_2$. The cells were then co-transfected at a 1:10 or 1:100 ratio of the pTwist-DENV2C-nLuc plasmid (expressing C-nLuc) with the HaloTag-ubiquitin plasmid (expressing HT-Ub, Promega). The transfection was performed using XtremeGeneHP DNA transfection reagent (Roche, #06366244001) according to the manufacturer's protocol. The transfected cells were incubated for 20 h at 37 °C, washed with 1 mL of PBS, trypsinized, and resuspended in OPTI-MEM Reduced Serum Medium without phenol red (Thermo Fisher Scientific, #11058021) supplemented with 4% FBS at a density of $2.2 \times 10^5$ cells/mL. The cells were divided into two pools, and each pool was treated with either DMSO (vehicle solvent; 0.1% final concentration) or HaloTag Nano-BRET 618 Ligand (Promega, 100 nM final concentration). 36 μL cells were dispensed in a white 384-well plate (Corning, #3570). A 10× concentration of degraders in OPTI-MEM Reduced Serum Medium without phenol red (100 μM for a 10 μM final concentration) was prepared alongside an equivalent dilution of DMSO as a negative control, adding an equal volume of DMSO to OPTI-MEM Medium. From this solution, 4 μL was added to each well, and the plates were incubated at 37 °C for 24 h. A 5× solution or 100-fold dilution of stock NanoBRET Nano-Glo substrate was prepared in OPTI-MEM Reduced Serum Medium without FBS, and 10 μL was added to each well. The assay was read immediately on a CLARIOstar plate reader (BMG Labtech), collecting donor emission at 460 nm/20 nm range and acceptor emission at 618 nm with a 610 nm LP filter. The acceptor emission value was then divided by the donor emission value for each sample to generate raw NanoBRET ratio values (typically decimal values), which were then converted to milliBRET units (mBU; whole numbers) by multiplying each raw BRET value by 1000. Next, the mean NanoBRET ratio for each set of samples was determined. The no-ligand control mean was subtracted from the experimental mean for the corrected NanoBRET ratio to factor in donor-contributed background or bleedthrough. The mean corrected BRET ratio was plotted for each compound using GraphPad Prism 10, and an unpaired t-test was performed to compare quantitative data. Statistically significant differences between compound-treated and DMSO-treated samples are shown by asterisks in the figures ($> 0.05$ (ns), $\leq 0.05$ (*), $p \leq 0.01$ (**), $p \leq 0.001$ (***), $p \leq 0.0001$ (****)). Additionally, the following Eq. (1) generated Z-factors for DMSO samples from each set to gauge assay consistency. Z-factor = $1 − (3 \times SD\_experimental + 3 \times SD\_no\text{-ligand control})/(Mean MBU\_experimental − Mean MBU\_no\text{-ligand control})$ (1).

### Cellular thermal shift assay (CETSA)

To determine the potential of CRBN stabilization by the lead degraders, negative controls, and parental inhibitor ST148, Huh7.5 cells were washed with PBS, trypsinized, and resuspended at $2.5 \times 10^6$ cells/mL in sterile PBS supplemented with 1× Protease inhibitor for each compound treatment condition (DMSO, ST148 parental inhibitor, E3 ligand lenalidomide, RPG-01-139, RPG-01-139-BUMP, RPG-01-132, and RPG-01-132-BUMP). 10 μM of the indicated compound or DMSO at a 0.1% final concentration was added to the cells, mixed thoroughly, and incubated at 37 °C for 1 h while shaking at 200 rpm. Meanwhile, the Thermocycler was preheated to run a temperature gradient between 48 and 68 °C. Following this, 70 μL cells were aliquoted into 96 PCR well plates, heated to variable temperatures for 3 min, and then cooled to 10 °C for 3 min. The cell lysates were collected by five freeze–thaw cycles followed by centrifugation at $17,500 \times g$ (rotor radius 8 cm) for 20 min at 4 °C. The abundance of CRBN and GAPDH protein was then characterized by Western blot analysis.

To investigate C stabilization by the lead degraders, Huh7.5 cells were seeded at a density of $1 \times 10^6$ cells/well in six-well plates. The cells were incubated for 24 h at 37 °C under 5% $CO_2$. The cells were then infected with DENV2 New Guinea C (NGC) at MOI 1 for 1 h, washed to remove extracellular virus, and overlaid with DMEM medium supplemented with 2% FBS. The infected cells were then harvested at 24 h post-infection. 1 mL of infected cells is incubated with 10 μM of each compound or 0.1% DMSO at 37 °C for 1 h. Following this, 70 μL of cells were aliquoted into 96 PCR well plates, heated to variable temperatures for 3 min, and then cooled to 10 °C for 3 min. The cell lysates were collected by five freeze–thaw cycles followed by centrifugation at $17,500 \times g$ (rotor radius 8 cm) for 20 min at 4 °C. The abundance of C and GAPDH protein was then characterized by Western blot analysis.

### CRBN cellular engagement assay

HiBiT-BRD4$_{BD2}$ Jurkat cells were maintained in RPMI medium supplemented with 10% fetal bovine serum (FBS). Approximately $0.5 \times 10^5$ HiBiT-BRD4$_{BD2}$ Jurkat cells were seeded in 50 μL of growth medium per well in opaque white 384-well plates. Compounds were dispensed using a D300e Digital Dispenser (Tecan) from DMSO stock solutions to achieve the desired final concentrations. Following a 1 h incubation with test compounds, cells were treated with dBET6 (MedChemExpress, #HY-112588) at 400 nM final concentration. After an additional 6 h incubation, endogenous BRD4 levels were quantified using the Nano-Glo HiBiT Lytic Detection System (Promega, #N3040), according to the manufacturer's instructions.

### Antiviral assay

Huh7.5 or Huh7.5 CRBN$^{-/-}$ cells were seeded into 24-well plates at $5 \times 10^4$ cells/well in DMEM supplemented with nonessential amino acids, 10 mM HEPES, and 2% FBS and incubated for 24 h. Cells were infected with the virus (DENV1 WP74, DENV2 NGC, DENV2 16681, DENV3 THD3 or DENV4 TVP360) at an MOI of 0.5 for 1 h and washed with PBS to remove the extracellular virus. The infected cells were then treated with compounds at varying concentrations in DMEM supplemented with nonessential amino acids, 10 mM HEPES, and 2% FBS. At 24 h post-infection, the cell lysates were collected for analysis of protein abundance via Western blotting. The culture supernatants were collected for viral titer quantification via viral plaque assay.

The yield of infectious particles after single-cycle replication was measured. Briefly, to titer DENV1 WP74, DENV2 NGC, DENV2 16681, or DENV4 TVP360, BHK21 cells were seeded into 24-well plates at $1 \times 10^5$ cells/well in MEM-α supplemented with 10 mM HEPES, and 5% FBS. To titer DENV3 THD3, Vero cells were seeded into 24-well plates at $1.5 \times 10^5$ cells/well in DMEM supplemented with nonessential amino acids, 10 mM HEPES, and 2% FBS. Culture supernatants from dengue infections were ten-fold serially diluted in HyClone Earle's 1× balanced salt solution (Cytiva, #SH30029.02), and 100 μL of each dilution was added to the cells in duplicates, followed by incubation for 1 h at 37 °C. Subsequently, the cells were washed with PBS, and 500 μL of methylcellulose supplemented with MEM-α, 10 mM HEPES, and 5% FBS overlay was added to each well. Following incubation for 4-5 days at 37 °C, cells were fixed and stained overnight at room temperature with

1% crystal violet in formaldehyde and PBS. Subsequently, the crystal violet solution was appropriately discarded, and the plates were washed thoroughly with deionized water and air-dried. The plaques were then counted using a lightbox, and the antiviral $EC_{50}$, corresponding to the compound concentration that resulted in a 50% reduction in viral titer, was determined by nonlinear regression in GraphPad Prism 10.

## RT-qPCR analysis of viral genomic RNA

Huh7.5 cells were seeded in 24-well plates at a density of $5 \times 10^4$ cells/well in DMEM supplemented with non-essential amino acids, 10 mM HEPES, and 2% FBS and incubated for 24 h. Cells were then infected with DENV2 NGC at an MOI of 0.5 for 1 h, washed with PBS to remove residual inoculum, and treated with compounds at the indicated concentrations in the same medium. At 2 h and 8 h post-infection, cells were lysed using iScript™ RT-qPCR Sample Preparation Reagent (Bio-Rad, #1708898), and total RNA was isolated according to the manufacturer's instructions. cDNA was synthesized from each sample using the iScript™ cDNA Synthesis Kit (Bio-Rad, #1708890). Quantitative PCR was performed using 1 μl cDNA as template with iQ™ SYBR Green Supermix (Bio-Rad, #1708880) and DENV2 capsid-specific primers synthesized by IDT (forward 5′-GGCGTTCCTTCGTTTCCTAA-3′, reverse 5′-AGCATCCTTCCAATCTCTTTCC-3′). Reactions were run in three technical replicates on a CFX96 Real-Time PCR system (Bio-Rad) with the following cycling conditions: 95 °C for 3 min, followed by 40 cycles of 95 °C for 10 s and 61 °C for 30 s, and a melt-curve analysis from 50–95 °C in 0.5 °C increments. Relative RNA levels were quantified using the ΔΔCq method and normalized to β-actin controls.

## Analysis of the effects of compounds on the dengue virus subgenomic replicon by microscopy

293-T-REx-DVGZ cells, a gift from Ted Pierson (NIH), were seeded in 384-well black, clear-bottom plates (Corning, #3764) at a density of $8 \times 10^3$ cells/well in DMEM supplemented with non-essential amino acids, 10 mM HEPES, 10% tetracycline-approved FBS, 2 mM glutamine, 5 μg/mL blasticidin S, and 300 μg/mL zeocin, and incubated for 20 h. Cells were then stained with Hoechst 33342 (ApexBio, #A3472) at a final concentration of 1 μg/mL for 5 min at 37 °C. Following nuclear staining, the medium was removed and replaced with the same seeding medium, with FluoroBrite™ DMEM (Gibco, #A1896701) substituted for standard DMEM to enhance fluorescence imaging. DMSO-normalized compounds were dispensed directly into wells using a D300e Digital Dispenser (Tecan) from DMSO stock solutions to achieve the desired final concentrations. GFP and Hoechst fluorescence was recorded at intervals from 0 to 40 h post-treatment to monitor replicon activity under compound exposure using an Olympus CellTIRF microscope in epi-fluorescence mode (10× objective). During acquisition, temperature, humidity, and $CO_2$ content were controlled with a Tokai-Hit stage top incubator and warming box. Images were analyzed by MATLAB (Mathworks, 2024b), and the relevant custom code was deposited in Zenodo[66]. Cell segmentation was estimated by the Hoechst signal, and GFP was measured in the segmented areas. For each image GFP signal was estimated as the median of all cells subtracted by the median of the non-segmented area. All replicates were averaged together. Trends in the figure are the smoothed spline interpolation of data.

## Electron microscopy

Huh7.5 cells were seeded into 6-well plates at $3 \times 10^5$ cells/well in DMEM supplemented with nonessential amino acids, 10 mM HEPES, and 2% FBS and incubated for 24 h. Cells were infected with the DENV2 NGC at an MOI of 20 for 1 h and washed with PBS to remove the extracellular virus. Then, the infected cells were treated with either 10 μM ST148, 30 μM RPG-01-132, or DMSO in DMEM supplemented with nonessential amino acids, 10 mM HEPES, and 2% FBS.

At 24 h post-infection, the cells were fixed in 2% glutaraldehyde and 4% paraformaldehyde in 0.1 M sodium cacodylate for 1 h at room temperature, then stored at 4 °C. Following three washes, each comprising 10 min, in 0.1 M sodium cacodylate, post-fixation was performed using 2% osmium tetroxide ($OsO_4$) and 2.5% potassium ferricyanide in 0.1 M sodium cacodylate for 1.5 h each at room temperature. Samples were washed twice in $ddH_2O$ before staining with freshly prepared thiocarbohydrazide for 20 min, followed by a second $OsO_4$ treatment for 90 min at room temperature. Subsequently, two more washes of $ddH_2O$, each for 30 min, were given. En bloc staining was performed using 1% uranyl acetate overnight at 4 °C, followed by an additional 2 h incubation at 50 °C, followed by two washes of $ddH_2O$, each for 30 min. Walton's lead aspartate staining was carried out for 2 h at 50 °C, with $ddH_2O$ washes after the staining step. Cells were then scraped, pelleted at $4000 \times g$, embedded in 10% gelatin in sodium cacodylate buffer, and post-fixed in cold 2% glutaraldehyde and 4% paraformaldehyde in 0.1 M sodium cacodylate. Dehydration was performed in a graded ethanol series (30%, 50%, 70%, 95%, and 100%) followed by acetonitrile. Resin infiltration was conducted using increasing concentrations of Embed 812-Hard in acetonitrile, culminating in 100% resin infiltration. Samples were embedded in fresh Epon resin and polymerized overnight at 60 °C. After curing, the samples were removed from molds and prepared for ultrathin sectioning using an ultramicrotome. The sections were imaged with a JEOL JEM-1400 transmission electron microscope operated at 120 kV and equipped with a Gatan OneView 4k × 4k sCMOS camera. (Note: our EM sample preparation protocol differed from standard protocols such as Scaturro et al. (J. Virol. 2014)[52], as we fixed cells as pellets and sectioned blocks rather than coverslips, resulting in less regular but still evident virion arrays.)

## IFN-β promoter activity assay

To determine the extent of degradation of transiently expressed C-nLuc, HEK293 cells were seeded in a white 96-well plate (Corning, #3917) at $8 \times 10^3$ cells/well in DMEM supplemented with nonessential amino acids, 10 mM HEPES, and 2% FBS and incubated for 24 h. After 24 h, the pTwist-DENV2C-nLuc plasmid (expressing C-nLuc) at 25 ng/well was transfected using XtremeGeneHP DNA transfection reagent (Roche, #06366244001). Immediately after the transfection, RPG-01-132 was added to the wells in three to six technical replicates in varying concentrations, with a 0.1% final DMSO concentration as a negative control. Based on the degradation profile, 0.625, 0.312, and 0.156 μM concentrations were chosen for the following assay.

To determine whether RPG-01-132 can rescue the inhibition of IFN-β promoter caused by dengue C protein, HEK293 cells were seeded in a white 96-well plate (Corning, #3917) at $8 \times 10^3$ cells/well in DMEM supplemented with nonessential amino acids, 10 mM HEPES, and 2% FBS and incubated for 24 h. pIFN-B-Luc plasmid (a gift from Jae Jung, Case Western Reserve University) was transfected at 6.3 ng/well using XtremeGeneHP DNA transfection reagent according to the manufacturer's protocol. After another 24 h of allowing cells to recover, the pTwist-DENV2C-nLuc plasmid (expressing C-nLuc) at 25 ng/well and high molecular weight polyinosinic:polycytidylic Acid [Poly(I:C), InvivoGen, #tlrl-pic] at 7.8 ng/well were co-transfected using XtremeGeneHP DNA transfection reagent. Immediately after the second transfection, ST148, RPG-01-132, or RPG-01-132-BUMP were added to the cells in eight replicates at 0.625, 0.312, and 0.156 μM with an equivalent mass of DMSO added to the cells as a vehicle control. 24 h after poly(I:C) transfection, the firefly luciferase signal was read on a CLARIOstar plate reader (BMG Labtech) after adding ONE-Glo EX Luciferase Assay reagent (Promega, #E8120) to the cells according to the manufacturer's protocol. Subsequently, NanoDLR Stop and Glo reagent (Promega, #N1610) were added to the wells at recommended volumes to quench the firefly Luc signal and read the NanoLuc signal on the same plate reader. The data were plotted at independent replicates using GraphPad Prism 10.

## Cell viability assay

Huh7.5 cells were seeded in a white 96-well plate at $1 \times 10^4$ cells/well in DMEM supplemented with nonessential amino acids, 10 mM HEPES, and 2% FBS and incubated for 24 h. Then, compounds were added to achieve the indicated concentrations and incubated for a further 24 h. After incubation, cellular ATP content was measured using CellTiter-Glo Luminescent Cell Viability Assay (Promega, #G7572). $CC_{50}$ values were determined using a nonlinear regression curve fit using GraphPad Prism 10. Each set of triplicate measurements in a given experiment was used for curve fitting; the $CC_{50}$ was calculated from $n = 2$ independent experiments.

## Proteomics

MOLT4 cells were utilized for these studies to enable sensitive detection of degradation of the IKZF proteins targeted by CRBN ligands, as detection of activity against these potential "off-targets" is challenging in Huh7.5 due to low expression. MOLT4 cells were treated with DMSO or 5 μM test compounds (RPG-01-132 or RPG-01-139) for 8 h. Cells were harvested by centrifugation and washed with phosphate-buffered saline (PBS) before snap freezing in liquid nitrogen. Cells were lysed by the addition of lysis buffer (8 M Urea, 50 mM NaCl, 50 mM 4-(2-hydroxyethyl)-1-piperazineethanesulfonic acid (EPPS), pH 8.5, Protease and Phosphatase inhibitors) and homogenization by bead beating (BioSpec) for three repeats of 30 s at 2400 strokes/min. The Bradford assay was used to determine the final protein concentration in the clarified cell lysate. 50 μg of protein for each sample was reduced, alkylated and precipitated using methanol/chloroform as previously described[67], and the resulting washed precipitated protein was allowed to air dry. Precipitated protein was resuspended in 4 M urea, 50 mM HEPES, pH 7.4, followed by dilution to 1 M urea with the addition of 200 mM EPPS, pH 8. Proteins were digested with the addition of LysC (1:50; enzyme:protein) and trypsin (1:50; enzyme:protein) for 12 h at 37 °C. Sample digests were acidified with formic acid to a pH of 2-3 before desalting using C18 solid phase extraction plates (SOLA, Thermo Fisher Scientific). Desalted peptides were dried in a vacuum-centrifuged and reconstituted in 0.1% formic acid for liquid chromatography-mass spectrometry analysis.

Data were collected using a TimsTOF Pro2 (Bruker Daltonics, Bremen, Germany) coupled to a nanoElute LC pump (Bruker Daltonics, Bremen, Germany) via a CaptiveSpray nano-electrospray source. Peptides were separated on a reversed-phase C18 column (25 cm×75 μm ID, 1.6 μM, IonOpticks, Australia) containing an integrated captive spray emitter. Peptides were separated using a 50 min gradient of 2%–30% buffer B (acetonitrile in 0.1% formic acid) with a flow rate of 250 nL/min and column temperature maintained at 50 °C.

Data were collected using a diaPASEF acquisition method where the precursor distribution in the DDA *m/z*-ion mobility plane was used to design an acquisition scheme for Data-independent acquisition (DIA) data collection, which included two windows in each 50 ms diaPASEF scan. Data was acquired using sixteen of these 25 Da precursor double window scans (creating 32 windows,) which covered the diagonal scan line for doubly and triply charged precursors, with singly charged precursors able to be excluded by their position in the m/z-ion mobility plane. These precursor isolation windows were defined between 400–1200 m/z and 1/k₀ of 0.7–1.3 V·s·cm⁻².

## LC-MS data analysis

The diaPASEF raw file processing and controlling peptide and protein level false discovery rates, assembling proteins from peptides, and protein quantification from peptides were performed using library-free analysis in DIA-NN 1.8[68] searched against a SwissProt human database (January 2021). Database search criteria largely followed the default settings for directDIA, including: tryptic with two missed cleavages, carbamidomethylation of cysteine, and oxidation of methionine and precursor Q-value (FDR) cut-off of 0.01. The precursor

quantification strategy was set to Robust LC (high accuracy) with RT-dependent cross-run normalization. Proteins with a low sum of abundance (<2000 × no. of treatments) were excluded from further analysis, and the resulting data were filtered to only include proteins that had a minimum of 3 counts in at least 4 replicates of each independent comparison of treatment sample to the DMSO control. Proteins with missing values were imputed by random selection from a Gaussian distribution either with a mean of the non-missing values for that treatment group or with a mean equal to the median of the background (in cases when all values for a treatment group are missing)[69] using in-house scripts in the R framework (R Development Core Team, R Foundation for Statistical Computing, 2014). Significant changes comparing the relative protein abundance of these treatments to DMSO control comparisons were assessed by a two-sided moderated *t*-test as implemented in the limma package within the R framework[70].

## GSPT1 DC₅₀ determination

Concentration-response data were fit in GraphPad Prism using the "absolute IC₅₀" function derived from a 4-parameter logistic model fit. For CC-90009, the dose range (10 μM–10 nM) did not fully capture the low-concentration plateau. Consequently, the reported DC50 values of 4 nM for CC-90009 and 13nM for SJ6986 represent extrapolated estimates that represent the upper bound of the DC50 since we achieved degradation of ≥50% of GSPT1 at the lowest concentration tested (10 nM).

## Cell permeability assay

The cell permeability of the degraders, negative controls, and the parental inhibitor ST148 was performed by WuXi AppTec Co., Ltd. MDR1-MDCK II cells (obtained from Piet Borst at the Netherlands Cancer Institute) were seeded onto Polycarbonate membranes (PC) in 96-well insert systems at $3.33 \times 10^5$ cells/mL until a confluent cell monolayer had formed (4–7 days). Test compounds were diluted with the transport buffer (HBSS with 10 mM HEPES, pH 7.4) from DMSO stock solution to a concentration of 2 μM (DMSO < 1.0%) and applied to the apical or basolateral side of the cell monolayer. Permeation of the test compounds from A to B direction or B to A direction was determined in duplicate. Digoxin was tested at 10 μM from A to B direction or B to A direction as well, while nadolol and metoprolol were tested at 2 μM in A to B direction in duplicate. The plate was incubated for 2.5 h in a $CO_2$ incubator at 37 ± 1 °C, with 5% $CO_2$ at saturated humidity without shaking. In addition, the efflux ratio of each compound was also determined. Test and reference compounds were quantified by LC-MS/MS analysis based on the peak area ratio of analyte/IS.

After the transport assay, the Lucifer yellow rejection assay was applied to determine the cell monolayer integrity. Buffers were removed from both apical and basolateral chambers, followed by the addition of 75 μL of 100 μM lucifer yellow in transport buffer and 250 μL transport buffer in apical and basolateral chambers, respectively. The plate was incubated for 30 min at 37 °C with 5% CO2 and 95.0% relative humidity without shaking. After 30 min incubation, 20 μL of the lucifer yellow samples were taken from the apical sides, followed by the addition of 60 μL of Transport Buffer. And then, 80 μL of lucifer yellow samples were taken from the basolateral sides. The relative fluorescence unit (RFU) of lucifer yellow was measured at 425/528 nm (excitation/emission) with an Envision plate reader.

The apparent permeability coefficient, $P_{app}$ (cm s⁻¹), was calculated using Eq. (1):

$$P_{app} = \frac{dC_r/dt \times V_r}{A \times C_0} \tag{1}$$

where $dC_r/dt$ is the change in compound concentration in the receiver chamber as a function of time (μM s⁻¹), $V_r$ is the solution volume in the

receiver chamber (0.075 mL on the apical side and 0.25 mL on the basolateral side), $A$ is the surface area for transport (0.143 cm$^2$), and $C_0$ is the initial concentration in the donor chamber (μM).

The efflux ratio was calculated using Eq. (2):

$$\text{Efflux ratio} = \frac{P_{\text{app}}(\text{BA})}{P_{\text{app}}(\text{AB})} \qquad (2)$$

Percent recovery was calculated using Eq. (3):

$$\% \text{ Recovery} = 100 \times \frac{(V_r \times C_r) + (V_d \times C_d)}{V_d \times C_0} \qquad (3)$$

where $V^d$ is the volume in the donor chamber (0.075 mL on the apical side and 0.25 mL on the basolateral side), and $C^d$ and $C^r$ are the final concentrations of compound in the donor and receiver chambers, respectively.

### Reporting summary
Further information on research design is available in the Nature Portfolio Reporting Summary linked to this article.

## Data availability
Source data are provided with this paper. The raw proteomics data generated in this study have been deposited in the PRIDE repository (ProteomeXchange Consortium) under accession code PXD063557. All quantitative data underlying the figures and Supplementary Figs. are provided in the accompanying Source Data file. Source data are provided with this paper.

## Code availability
The MATLAB code used to analyze the dengue subgenomic replicon microscopy data in this study has been deposited in Zenodo and is publicly available at https://doi.org/10.5281/zenodo.18259090.

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

## Acknowledgements

This work was supported by NIH Award R01 AI148632 awarded to P.L.Y. and N.S.G. We thank the following colleagues for sharing reagents: Charles Rice for Huh7.5 cells, Trung Pham for HEK293 cells, Eva Harris for BHK-21 cells; Ted Pierson for 293-T-REx-DVGZ cells; Aravinda De Silva for DENV1 WP74, DENV3 THD3, and DENV4 TVP360 viruses; Lee Gehrke for DENV2 New Guinea C virus; Karla Kirkegaard for DENV2 16681 wild-type and S34L viruses; John Aaskov for DENV2 C monoclonal antibody;

Aaron Schmidt for DENV2 E antibody; and Jan Carette for guide RNAs to generate CRBN knockout cells. We thank John Perrino from the Cell Sciences Imaging Facility at Beckman Center, Stanford University, for helping us with the sectioning of resin-embedded cells for electron microscopy. We thank members of the Yang and Gray laboratories for helpful discussions. Schematics in Figs. 1a, 2a, 3a, e, and 5b, and Supplementary Figs. 4a–c, 8a, 12a, and 13a were created using BioRender and are published here under a CC-BY-NC-ND license. Open access to this article does not include using the images created in BioRender.

## Author contributions

P.L.Y. conceived the study. A.C., K.A.D., O.A., E.S.F., N.S.G., and P.L.Y. designed the experiments. A.C. led the biological evaluation experiments; designed, performed, and optimized virology and biochemical assays; and analyzed and interpreted the data. L.N.W., R.P.G., and Z.L. performed chemical synthesis. K.A.D. performed the MS proteomics experiments. O.A. performed the competition-based E3 engagement assay, acquired the microscopic images for the subgenomic replicon experiment, and analyzed the data. Y.Z. prepared the EM samples and acquired the electron micrographs. A.C., L.N.W., and P.L.Y. wrote the paper with input from all authors.

## Competing interests

P.L.Y. is a founder and science advisory board member (SAB) of Viraccio Therapeutics. N.S.G. is a founder, science advisory board member (SAB), and equity holder in Syros, C4, Allorion, Lighthorse, Inception, Matchpoint, Shenandoah (board member), Larkspur (board member), and Soltego (board member). The Gray lab receives or has received research funding from Novartis, Takeda, Astellas, Taiho, Jansen, Kinogen, Arbella, Deerfield, Springworks, Interline and Sanofi. K.A.D. receives or has received consulting fees from Neomorph Inc. and Kronos Bio. E.S.F. is a founder, SAB and equity holder in Neomorph (board member), Civetta, Anvia (board member), Proximity, and Stelexis; consultant/SAB and equity holder in Ajax, Avilar, and Photys; equity holder in Lighthorse and CPD4 (board member). The Fischer lab has received or continues to receive research funding from Novartis, Astellas, Ajax, Deerfield, Springworks, and Interline. The remaining authors declare no competing interests.
