## [Transparent Peer Review file · Nature Communications]

Degraders of the dengue virus capsid protein exhibit differentiated pharmacology relative to capsid inhibitors

Corresponding Author: Professor Priscilla Yang

Version 0:

Reviewer comments:

Reviewer #1

(Remarks to the Author)

In this study, the authors describe the development of RPG-01-132, a dengue virus (DENV) capsid-targeting PROTAC molecule utilizing the established capsid inhibitor ST148 as a targeting ligand. The compound demonstrates promising pan-serotype activity against all four DENV serotypes and maintains efficacy against an ST148-resistant viral strain. Mechanistic investigations reveal sub-micromolar antiviral potency coupled with CRBN-mediated capsid protein degradation, achieving dual antiviral effects through both virion assembly inhibition and restoration of host interferon response. While the conceptual framework and preliminary findings present notable scientific interest, and the majority of experiments appear technically sound, the manuscript would benefit from enhanced experimental depth, more rigorous data interpretation, and improved structural clarity. Several critical limitations currently prevent definitive validation of the PROTAC mechanism hypothesis:

1. Insufficient Validation of Target Engagement

The current evidence for ternary complex formation between DENV capsid (C), CRBN E3 ligase, and RPG-01-132 remains circumstantial. While leveraging known ligands (ST148 and CRBN binder) provides theoretical rationale, direct experimental confirmation is essential. Recommended validation approaches include: In vitro binding assays (SPR/ITC) demonstrating compound interaction with both targets; co-immunoprecipitation and/or co-localization imaging demonstrating ternary complex formation. The current CETSA data may reflect indirect stabilization effects rather than direct target engagement.

2. Incomplete Degradation Pathway Characterization

Key proteasome-dependent degradation controls are missing from current data: For example: Quantitative PCR analysis of capsid mRNA levels under treatment conditions; Cycloheximide chase experiments to measure protein half-life; Proteasome inhibitor (MG132/epoxomicin) rescue experiments. These controls are essential to distinguish true degradation from transcriptional regulation or translational inhibition.

3. Discrepancies in Hook Effect Observations

The reported hook effect requires clarification given conflicting evidence: Absence of characteristic hook effect in main antiviral assays (up to 20 μM); Putative stabilization observed in C-nLuc reporter system ($>1 \mu\text{M}$). This discrepancy should be resolved through parallel dose-response comparisons in both systems, potentially including extended concentration ranges and orthogonal detection methods.

4. Lack of Genetic Validation

A critical gap exists in functional validation: No degrader-resistant capsid mutants presented to establish mechanism. Missing viral replication competence assays with engineered C mutants. Generation of resistance mutations through directed evolution or structure-guided mutagenesis would provide compelling loss-of-function evidence.

While the PROTAC-based antiviral strategy presents an innovative approach to combatting dengue infection, the current evidence remains insufficient to conclusively establish RPG-01-132's mechanism of action as true targeted protein degradation. Addressing these experimental gaps through additional target engagement studies, degradation pathway validation, and genetic evidence would significantly strengthen the mechanistic claims and enhance the manuscript's impact.

Reviewer #2

(Remarks to the Author)

In this study, the authors presented the development of PROTAC degraders against the dengue virus capsid protein as a potential new therapeutic strategy for the treatment of the dengue virus infection. Two series of PROTAC dengue virus (DENV) capsid degraders were designed using a known capsid inhibitor ST148 as a targeting ligand and cereblon and VHL-1 ligands. The authors identified two PROTAC DENV capsid degraders, namely RPG-01-132 and RPG-01-139 for further investigation. Their data showed that RPG-01-132 and RPG-01-139 induced degradation DENV capsid via a PROTAC mechanism. Furthermore, the authors further showed that RPG-01-132 inhibited multiple functions of the dengue capsid protein and demonstrated a broad activity against dengue virions, as compared to ST148. Overall, this is a good proof-of-concept study that induced degradation of the dengue virus capsid protein could be a potential therapeutic strategy for the treatment of the dengue virus infection.

Some revisions are recommended. First, RPG-01-132 (and also RPG-01-139) induced degradation of GSPT1 protein, which may contribute to the activity of RPG-01-132 on inhibition of multiple functions of the dengue capsid protein and its broad activity against dengue virions. In the study, the authors compared the potencies RPG-01-132 and RPG-01-139 vs two known and very potent GSPT1 inhibitors and concluded that because RPG-01-132 and RPG-01-139 only induced degradation of GSPT1 at low micromolar concentrations and were much weaker than those two known GSPT1 degraders, their GSPT1 degradation activity did not contribute to their activity on inhibition of multiple functions of the dengue capsid protein and its broad activity against dengue virions. However, RPG-01-132 was not a very potent degrader and inhibitor of the dengue virus capsid protein and also had a micromolar potency against dengue virions. Hence, it is very likely that its GSPT1 degradation activity of RPG-01-132 contributes to the degradation of the dengue capsid protein, as well as its activity against dengue virions.

Second, as compared to many reported PROTAC degraders, the degradation potency (DC50) and efficiency (Dmax) are very modest. While this study provided a proof-of-concept, the therapeutic application of a PROTAC degrader of the dengue virus capsid protein will be very limited. Please discuss this important limitation in the paper.

Reviewer #3

(Remarks to the Author)

In this study, Chakravarty and colleagues report a novel class of dengue virus inhibitors which seemingly induce the degradation of the capsid viral protein (C). Following screening and characterization, the authors focus the study on their best compound, RPG-01-132, which possesses an antiviral potency in the micromolar range for dengue virus serotype 2 based on the determination of the EC50 value. Notably using KO cells and control compounds, the authors convincingly demonstrate that the decrease in C expression in DENV-infected cells is dependent on CRBN.

Although, this approach is elegant and make the proof-of-concept that it can identify novel classes of antivirals in principle, the antiviral potency of the drug candidates was rather low. Indeed, the EC50 of RPG-01-132 was around 1 μ M against DENV serotype 2 while the maximum reduction of viral titers in treated infected cells was less than 10-fold. This was in fact lower than the phenotype observed with ST148, the (non-degrader) capsid inhibitor whose backbone was used to generate the studied compounds. From a drug development point of view, such activity is not sufficient, especially since the lead compound has a very poor antiviral activity against dengue viruses from serotypes 1,3, and 4 while it was not challenged in in vivo infection models. It is interesting that it is still active against an ST148-resistant DENV2 strain; but this does not preclude that the virus will adapt rapidly to RPG-01-132, even with compensatory mutations simply increasing replication.

The authors claim that the activity of the RPG-01-132 is capsid-specific. It is indeed quite clear that in drug-treated infected cells, C levels decrease in a drug-concentration- and CRBN-dependent manner. However, in infected cells, the expression of C as well as that of all viral proteins is directly dependent on the abundance of viral RNA, and thus on the efficiency of RNA replication. Considering this, if the drug targets the RNA replication step of the life cycle (instead of C as concluded by the authors), it is expected that this will result in the decrease in viral protein expression independently of C. Thus, to prove that the drug mode-of-action is C-dependent/specific, the authors must evaluate the impact of drug treatment on: 1- the expression of other structural and nonstructural viral proteins using western blotting (which should not change if the hypothesis is correct), 2- the levels of viral RNA (to show that replication is not inhibited), and 3- the levels of untagged C, when expressed alone in a replication-independent context. Measuring C expression in a luciferase assay using a tagged protein is not appropriate, especially since the data of Fig S10 are misleading. With this indirect readout, protein expression seems to increase upon drug treatment, which is the opposite of what is expected.

Even if the two drugs are structurally related, the mode-of-action of ST148 and RPG-01-132 are different. Given that, as explained above, the C-specificity of RPG-01-132 is not addressed, the authors must confirm that this drug is targeting the C-dependent steps of the life cycles, namely viral assembly and entry/uncoating. They should test the drug on several assays

such as time-of-addition assays (to test entry), compare RNA levels and the production of infectious particles in infected cells to ensure that the decrease in titers is not due to a reduction of replication, and confirm that the drug does not impact the replication of a DENV sub-genomic replicon (which does not express C and does not make viral particles).

All western blots need to be quantified to help the reader to evaluate the phenotypes. This is especially true for Fig 2 (the starting point of the study) and Fig S3, which as such is not convincing. Indeed, in this figure, changes in C and CRBN stability is really not obvious since the reference protein levels (100%) at the lowest temperature (48°C) change among drug treatments.

Figure 5 is not convincing at all with the provided image quality. As such, it is hard for the reader to evaluate that the arrows indicate virion as stated. They do not look like the previously reported regular arrays. Magnified images are required to appreciate this. Moreover, the indicated viral factories (VF) do not resemble the endoplasmic reticulum-derived invaginated vesicles typically induced by DENV. One could argue that these structures are instead reminiscent of autophagosomes and not VFs. More importantly, why VFs (which host RNA replication) would be impacted by the drug treatment if the compound targets viral assembly. Alternatively, if the drug targets replication, it is expected that less VFs will be observed, simply because there are less viral RNA and viral proteins, and not because VF biogenesis is directly targeted by the drug. This needs to be rationalized.

Additional comments:

- What is the impact of the treatment of all the drugs of Fig 1 on viral titers. This would be more informative for an antiviral potency than C expression
- Fig S4: Although this is inferred, what FKBP12 western blot control for. This is not indicated.
- Why was the proteomic study performed in MOLT4 cells while the rest of the study was done exclusively in Huh7.5 cells?
- It is unclear why the authors state that the increase in lysosomes abundance is indicative of an aborted replication process. It could simply reflect that the autophagy is induced, which is not per se an antiviral process.
- Fig S6: It would be helpful that the BRET assay principle is explained in more details. A schematic could be relevant.
- What is the impact SJ6986 treatment on DENV titers?
- Fig 5C: in the Y-axis, it should be FLU and not RLU
- What is the impact of ST148 treatment in the IFN β induction assay?

Version 1:

Reviewer comments:

Reviewer #1

(Remarks to the Author)

The authors have satisfactorily addressed most of the questions and suggestions from the previous review, resulting in an improved manuscript. For further strengthening, I would suggest performing a statistical analysis of the results shown in Supplementary Figure 8 (NanoBRET assay), as the observed change in BRET value appears modest. Additionally, including a negative control—such as an irrelevant donor/acceptor pair—would help substantiate the conclusion.

Reviewer #2

(Remarks to the Author)

The authors have done extensive new experiments to address all the concerns raised by three reviewers, including those points raised in my previous review. On balance, although RPG-01-132 is still a modestly potent PROTAC degrader of DENV capsid, the study nevertheless has provided high-quality of data to demonstrate the proof-of-concept for this compound and the overall direction. Importantly, the authors have included discussions to clearly state the limitations of RPG-01-132 and future directions. Therefore, I support the publication of this revised manuscript in Nature Communications.

Reviewer #3

(Remarks to the Author)

In this revised version of their manuscript, the authors have substantially improved the study and addressed many of my important comments. They convincingly demonstrate that the mode-of-action of the drug is capsid-specific in infected cells and does not involve viral replication, translation or entry (to some extent in Fig S12).

However, while replication and translation can be ruled out, the precise step targeted by the drug remains unclear. It is likely that assembly is targeted but this is not unambiguously shown as such. Evidence of a decreased viral particle abundance is provided with electron microscopy data, but these observational phenotypes are not quantified. This is challenging to do. Thus, in order to support the conclusion, the authors should use the same experimental set-up as in Fig 2B, and measure in parallel intracellular and extracellular infectivity (by plaque assays) as well as viral intracellular and extracellular RNA (by RT-qPCR). If assembly is impaired, intracellular infectivity is expected to decrease while intracellular RNA (at this 24 hpi time point) should remain constant (further validating the absence of impact on RNA synthesis/translation). The authors have assessed viral RNA levels at 2 and 8 hours (Fig S12) but this should also be done at a time point at which the drug phenotype is observed, that is 24 hpi.

New Fig. S17 supports that the drug induces a decrease in capsid levels when expressed alone. However, the phenotype is rather modest (50% decrease at best for only one drug concentration) as compared to infection conditions (Fig 3) while it increased up to 10-fold at high concentration. Considering these mixed phenotypes, to unambiguously confirm that C is degraded upon drug treatment, similar experiment (at 0.62 μ M) should be repeated in MG132-treated cells or in CRBN KO

cells, in which C levels should be rescued.

Minor comment: In the description of RT-qPCR assays, it is not appropriate to use the term capsid RNA which is in fact the viral (genomic) RNA. The way it is written implies that there are two different RNA species. There is only one viral protein-encoding RNA which is translating into the viral polyprotein. This must be corrected.

We have provided point-by-point responses (black text) to all Reviewer comments (blue text) below. We numbered the Reviewer comments to facilitate referencing critiques and responses across this document. New data incorporated into the manuscript and Supplementary Information are also reproduced here in the point-by-point response. Manuscript and Supplementary files are also provided with all changes to the text in red.

Reviewer #1 (Remarks to the Author):

In this study, the authors describe the development of RPG-01-132, a dengue virus (DENV) capsid-targeting PROTAC molecule utilizing the established capsid inhibitor ST148 as a targeting ligand. The compound demonstrates promising pan-serotype activity against all four DENV serotypes and maintains efficacy against an ST148-resistant viral strain. Mechanistic investigations reveal sub-micromolar antiviral potency coupled with CRBN-mediated capsid protein degradation, achieving dual antiviral effects through both virion assembly inhibition and restoration of host interferon response. While the conceptual framework and preliminary findings present notable scientific interest, and the majority of experiments appear technically sound, the manuscript would benefit from enhanced experimental depth, more rigorous data interpretation, and improved structural clarity. Several critical limitations currently prevent definitive validation of the PROTAC mechanism hypothesis:

1. Insufficient Validation of Target Engagement

The current evidence for ternary complex formation between DENV capsid (C), CRBN E3 ligase, and RPG-01-132 remains circumstantial. While leveraging known ligands (ST148 and CRBN binder) provides theoretical rationale, direct experimental confirmation is essential. Recommended validation approaches include: In vitro binding assays (SPR/ITC) demonstrating compound interaction with both targets; co-immunoprecipitation and/or co-localization imaging demonstrating ternary complex formation. The current CETSA data may reflect indirect stabilization effects rather than direct target engagement.

Response: To address this critique, we utilized a NanoBRET® assay, an approach well-validated in the TPD field for detection of ternary complex formation *in cellulo*¹⁻⁴. Briefly, we made use of a commercially available NanoBRET® system (Promega NanoBRET Ternary Complex Assay) and ectopically expressed the capsid protein as a fusion with nanoluciferase (C-nLuc) alongside a HaloTag-CRBN protein. Formation of a ternary complex by HaloTag-CRBN, RPG-01-132, and C-nLuc results in energy transfer between the donor (HaloTag-CRBN) and acceptor (C-nLuc) and an increase in the NanoBRET® signal upon addition of the NanoBRET Nano-Glo substrate and NanoBRET 618 ligand. As shown below (and in **Supplementary Fig. 8a-d**), RPG-01-132 caused a concentration-dependent increase in the BRET ratio at both 1:10 and 1:100 donor-to-acceptor ratios across multiple replicate experiments, consistent with the formation of a ternary complex by CRBN, RPG-01-132, and DENV capsid. We can moreover infer that these ternary complexes are competent for targeted protein degradation since the C-nLuc donor luminescence increases when the proteasome inhibitor MG132 is included with RPG-01-132 (**Supplementary Fig. 8e-g**). These NanoBRET data provide direct, quantitative evidence of ternary complex formation between CRBN, DENV capsid, and RPG-01-132, and MG-132 rescue of donor signal shows that the complex is productive for downstream degradation.

Supplementary Figure 8. Detection of ternary complex formed by RPG-01-132 with CRBN and DENV capsid in live cells. (a) Schematic of NanoBRET® assay used to detect induced proximity of CRBN or ubiquitin with DENV capsid protein. Created in BioRender.com. (b-d) Ternary complex formed by DENV2 C-nLuc (donor), HaloTag-CRBN (acceptor), and RPG-01-132 was assessed at donor: acceptor ratios of 1:10 (replicates I and II) and 1:100 (replicate II). Raw NanoBRET ratios were calculated as acceptor/donor emission, multiplied by 1000 to obtain mBU, and then corrected by subtraction of the values observed for the no-NanoBRET618 ligand control. RPG-01-132 induced a concentration-dependent increase in corrected BRET ratio relative to the DMSO control, consistent with ligand-driven ternary complex formation. (e-g) The ternary complexes observed in this system are functional for

targeted protein degradation, as evidenced by the rescue of luminescence produced by C-nLuc proteasome inhibitor MG-132 (10 μ M) is included. Samples in **e-g** are from the same experiments depicted in **b-d**. Data shown as mean \pm SD from technical replicates. I and II are $n = 2$ independent biological replicates.

The following text has been added to the main manuscript:

"As additional confirmation of the TPD mechanism, we conducted assays to detect ternary complex formation and ubiquitination of C and to monitor the effects of RPG-01-132 in these assays. We utilized a commercial NanoBRET assay to monitor ternary complex formation by CRBN, RPG-01-132, and an ectopically expressed C-nLuc fusion protein in HEK293 cells (**Supplementary Fig. 8a**). Cells co-transfected with C-nLuc and HaloTag-CRBN fusion protein (HT-CRBN) expression plasmids were treated with either DMSO or RPG-01-132 along with the NanoBRET 618 ligand for 4 hours, at which time a NanoBRET Nano-Glo substrate was added, and both donor (nLuc: 460 nm) and acceptor (HaloTag: 618 nm) signals were measured. Consistent with ternary complex formation, we observed concentration-dependent BRET signal in the presence of RPG-01-132 (**Supplementary Fig. 8b-d**). These ternary complexes appear competent for TPD, as evidenced by rescue of the C-nLuc luminescence with the inclusion of proteasome inhibitor MG-132 (**Supplementary Fig. 8e-g**). This was corroborated by NanoBRET-based detection of ubiquitinated C-nLuc in the presence of RPG-01-132, but not RPG-01-132-BUMP (**Supplementary Fig. 8a, 9**). Densitometric quantification of the blots in **Fig. 3b** is provided in **Supplementary Fig. 7**. These experiments demonstrate that RPG-01-132 forms a productive ternary complex with CRBN and DENV2 C and causes concentration-dependent depletion of DENV2 C via a mechanism that requires engagement of both CRBN and C and neddylation of CRL4^{CRBN} and proteasome activity. Taken together, they validate DENV C as a target for TPD and establish assays to enable development of DENV C degraders."

2. Incomplete Degradation Pathway Characterization

Key proteasome-dependent degradation controls are missing from current data: For example: Quantitative PCR analysis of capsid mRNA levels under treatment conditions; Cycloheximide chase experiments to measure protein half-life; Proteasome inhibitor (MG132/epoxomicin) rescue experiments. These controls are essential to distinguish true degradation from transcriptional regulation or translational inhibition.

Response: We appreciate the Reviewer's suggestion to include additional controls to demonstrate that the antiviral activity is due to capsid degradation and not due to other modes of action affecting viral RNA transcription, replication, or translation. This critique was echoed in part by Reviewer 3 (comments 3 and 4), who made other suggestions for additional experiments. Here, we present the experiments requested by Reviewer 1 first, followed by relevant experiments requested by Reviewer 3 that bear on the important points made by both Reviewers regarding the mechanism. As suggested by Reviewer 1, we performed quantitative RT-qPCR analyses of dengue virus capsid mRNA under the treatment conditions outlined in **Supplementary Fig. 12a**. Huh7.5 cells infected with DENV2 NGC strain at MOI 0.5 were treated with ST148 and RPG-01-132 or RPG-01-132-BUMP at concentrations corresponding to 5-times the antiviral EC₅₀ values. Total RNA was extracted, reverse-transcribed, and analyzed by qPCR using primers specific for DENV C. As shown in **Supplementary Fig. 12b**, no significant changes in capsid mRNA abundance were observed compared to the vehicle control. To further rule out that RPG-01-132 might be acting through effects on translation, transcription, or replication of viral RNA, we conducted experiments suggested by Reviewer 3 (comment 3) in a DENV2 subgenomic replicon system⁵ that lacks the viral structural genes (E, prM, C) and thus allows assessment of effects on viral translation and RNA replication in the absence of viral entry and assembly steps. Neither ST-148 nor RPG-01-132 exhibits activity in this system (**Supplementary Fig. 13**), arguing that the antiviral activity of these compounds

against live DENV is not due to effects on viral RNA transcription, replication, or translation but, rather, due to direct effects on the DENV capsid protein.

The experiments showing rescue of capsid protein in the presence of MG-132 (proteasome inhibitor) and MLN4924 (neddylation inhibitor) (**Supplementary Figs. 8e-g, Fig. 3**) alongside the data demonstrating ternary complex formation and capsid ubiquitination (**Supplementary Figs. 8, 9**) also support a targeted protein degradation mechanism for RPG-01-132 versus mechanisms with more general effects of the compound on viral or host transcription or translation. Based on these findings, and with all respect to the Reviewer 1, we elected not to measure protein half-life in a cycloheximide chase experiment since the subgenomic replicon and viral RNA quantification alongside the experiments described in our response to Reviewer 3, Comment 3 collectively provide direct and rigorous evidence that the loss of capsid protein and the antiviral activity arise from a targeted protein degradation mechanism.

a

b

Supplementary Figure 12. Determination of RPG-01-132's effect on steady-state abundance of viral RNA. (a) Schematic of the experiment (created in BioRender). Huh7.5 cells infected with DENV2 NGC strain (at MOI 0.5) were treated with RPG-01-132 (2.5 μ M), its negative control RPG-01-132-BUMP (2.5 μ M), or parental inhibitor ST148 (1 μ M) at concentrations corresponding to five-fold greater than the antiviral EC₅₀ values for the respective compounds as established in **Fig. 4** and **Supplementary Fig. 6**. Following incubation for 2 h and 8 h, cells were harvested, and total RNA extracted for cDNA synthesis and subsequent qPCR using primers specific for the capsid gene. (b) RT-qPCR analysis of dengue virus capsid RNA abundance. No significant changes were observed in RPG-01-132 compared to the DMSO control, indicating that reductions in capsid protein and viral yield are not due to effects of the compounds on viral RNA. Representative data are from $n = 2$ independent experiments.

Supplementary Figure 13. Effect of RPG-01-132 in the DENV2 subgenomic replicon model. (a) Schematic of the subgenomic DENV2 replicon stably harbored by 293-TREx-DVGZ cells, a well-validated model for studying DENV translation, transcription, and RNA replication⁵. Accumulation of viral genomic RNA can be reliably monitored by detection of the eGFP reporter encoded on the replicon RNA. (b-e) 293-TREx-DVGZ cells were treated with RPG-01-132, RPG-01-132-BUMP (20 nM - 20 μM), ST148, or the NS4B inhibitor NITD-688 (10 nM - 20 μM). GFP fluorescence, which is proportionate to the abundance of replicon RNA, was measured from 0-40 h after compound addition. NITD688 potentially inhibited replicon replication, validating the assay. In contrast, RPG-01-132, RPG-01-132-BUMP, and ST148 had no effect in this system, consistent with a capsid-specific mechanism rather than a more general effect on viral RNA translation, transcription, or replication. (f) Data from b-e collected at the 40h time point plotted versus compound concentration. (g,h) Western blot for NS4B to verify that the reduced eGFP reporter signal reflects inhibition of replication and a decrease in viral protein following 24-hour treatment with NITD688 and no change in NS4B abundance caused by RPG-01-132.

The following text has been added to the main manuscript to the addition of these data:

"Since the degrader antiviral activity is notably observed at concentrations that cause less than 50% depletion of intracellular C, this could suggest that even a modest depletion of C is sufficient to block the assembly of DENV virions. Alternatively, this might indicate that RPG-01-132 and RPG-01-139 exert antiviral activity through mechanisms other than loss of capsid function. For example, any mechanism

inhibiting replication of the viral RNA genome would also be expected to cause depletion of not only C but all viral proteins over time, since the template for translation of these proteins would be reduced. To test our hypothesis that RPG-01-132's antiviral activity is due to specific loss of C, we conducted additional experiments to rule out more general mechanisms. First, we reasoned that if RPG-01-132's antiviral activity is due to a more general effect on replication, this would be reflected in depletion of other DENV proteins and not only C. Experiments examining the effect of RPG-01-132 on the abundance of envelope (E), NS4B, and NS5 detected no changes (**Supplementary Fig. 11**), confirming that the depletion is specific for C and disfavoring the idea that RPG-01-132 has a general effect on viral replication or viral RNA. As another, more explicit test of whether RPG-01-132's antiviral activity is due to more general effects on viral replication, we examined whether RPG-01-132 changes the steady-state abundance of the viral genomic RNA and found no effect on C RNA abundance (**Supplementary Fig. 12**). We also corroborated that RPG-01-132 does not affect translation or replication of the viral RNA using a DENV2 subgenomic replicon system⁵, which lacks the viral structural genes, including C, and is a well-established model for studying DENV RNA translation and replication in the absence of the steps of viral entry and virion assembly. Whereas the NS4B inhibitor NITD688 caused a reduction in replicon activity, RPG-01-132 and ST148 had no effect in this model (**Supplementary Fig. 13**), consistent with the idea that their antiviral activities are capsid-specific. Together, these experiments rule out the possibility that RPG-01-132's antiviral activity is due to more general effects on DENV replication and support the hypothesis that its effect on DENV is due to specific loss of C protein."

3. Discrepancies in Hook Effect Observations

The reported hook effect requires clarification given conflicting evidence: Absence of characteristic hook effect in main antiviral assays (up to 20 μM); Putative stabilization observed in C-nLuc reporter system (>1 μM). This discrepancy should be resolved through parallel dose-response comparisons in both systems, potentially including extended concentration ranges and orthogonal detection methods.

Response: We appreciate the Reviewer's request to clarify the hook effect. After performing additional experiments and side-by-side analyses, we can now report that we observe a hook effect only in the overexpression reporter system, and not in the primary antiviral readouts (tested to 20 μM RPG-01-132). We believe that this divergence is mechanistically coherent with established degrader pharmacology and reflects the concentration-dependent balance between binary and ternary complexes. Hook effects at high heterobivalent degrader concentrations arise when binary complexes (degrader–target protein and degrader–E3) compete with ternary complex formation. Both modeling and experiments in the TPD field have shown that the occurrence of a hook effect depends on the concentrations of each component and cooperativity in the formation of the ternary complex and that differences in target protein abundance between experimental systems can shift the hook effect^{3,6-9}. In the case of our experiments, the ectopically expressed capsid protein is expressed at high levels compared to endogenous expression of capsid protein under the conditions of viral infection employed in our studies. In addition, in the natural infection, there is constant flux of capsid protein out of the cell due to assembly of new viral nucleocapsids and the budding and export of new virions. This mechanism for removing capsid protein is absent in the ectopic expression system, which contributes to a greater abundance of the target in this system compared to in our infection experiments.

4. Lack of Genetic Validation

A critical gap exists in functional validation: No degrader-resistant capsid mutants presented to establish mechanism. Missing viral replication competence assays with engineered C mutants. Generation of resistance mutations through directed evolution or structure-guided mutagenesis would provide compelling loss-of-function evidence.

Response: We appreciate the Reviewer's point on the use of genetic validation strategies. This was echoed in part by Reviewer 3's comment 2 so we refer Reviewers to our response to that comment as well. While we agree that resistance studies have historically been useful in DAA target validation, we, respectfully, do not think that these experiments are necessary in this case. First, it is well-established that the parental targeting ligand is DENV capsid specific, and we validated this in our hands by showing that the previously selected S34L resistance mutation confers ST148-resistance in our experiments (**Fig. 6, Supplementary Fig. 6**). While it is possible that the addition of the linker and E3 ligand could lead to off-targets not observed for ST148, we do not believe this to be the case based on the proteomics experiments, which revealed only GSPT1, a known target of the CRBN-binding ligand, as an additional target of RPG-01-132 (**Supplementary Fig. 14**). Second, we have multiple, independent lines of evidence that firmly establish a capsid-specific mechanism of action for RPG-01-132, including: (i) direct demonstration of CRBN-RPG-01-132-capsid ternary complex formation in live cells (**Supplementary Fig. 8**); (ii) ubiquitination of capsid specifically induced by RPG-01-132 (**Supplementary Fig. 9**); (iii) absence of antiviral activity in the subgenomic replicon model, which lacks capsid protein (**Supplementary Fig. 13**).

We also, respectfully, have not conducted experiments to select for or engineer mutants resistant to RPG-01-132. This is in large part because in our past development of degraders targeting hepatitis C virus NS3 and DENV envelope protein, we did not observe the emergence of a resistant quasispecies over > 10 passages. These negative results cannot prove that resistance against these antiviral degraders won't occur; however, we think the lack of resistant quasispecies emergence reflects the general resilience of antiviral degraders to single point mutations that decrease drug-binding. In the case of the DENV envelope degrader, we were able to begin to observe partial resistance with the introduction of three or more mutations to the compound-binding site; however, we benefitted in that case from knowing which mutations could reduce binding without disrupting E protein structure or function. We believe that attempting a similarly targeted approach to engineer resistance against RPG-01-132 goes beyond the scope of the current study since it's unclear what combination of substitutions can reduce ST148-binding but maintain viral replication, and since we do not think that we need a degrader-resistant mutant to support our conclusion that the antiviral activity of RPG-01-132 is due to activity against DENV capsid. We also note that conducting directed evolution or structure-guided mutagenesis to select for degrader-resistant capsid variants has become challenging for biosafety reasons since these types of studies are currently considered under NIH guidelines to be gain-of-function experiments.

5. While the PROTAC-based antiviral strategy presents an innovative approach to combatting dengue infection, the current evidence remains insufficient to conclusively establish RPG-01-132's mechanism of action as true targeted protein degradation. Addressing these experimental gaps through additional target engagement studies, degradation pathway validation, and genetic evidence would significantly strengthen the mechanistic claims and enhance the manuscript's impact.

Again, we thank Reviewer 1 for the critiques that prompted us to conduct additional experiments to support our conclusions. In summary, we have added new data showing ternary complex formation and demonstrating rescue of capsid degradation by proteasome inhibitor MG132 (**Supplementary Fig. 8**) and demonstrating no effect of RPG-01-132 on DENV RNA by RT-qPCR (**Supplementary Fig. 12a, b**) or on DENV translation or RNA replication in the subgenomic replicon assay (**Supplementary Fig. 13**). We have also modified the manuscript text in numerous places, as noted, to provide a clearer context for our experiments and to be more scrupulous in our interpretations and conclusions.

Reviewer #2 (Remarks to the Author):

In this study, the authors presented the development of PROTAC degraders against the dengue virus capsid protein as a potential new therapeutic strategy for the treatment of the dengue virus infection. Two series of PROTAC dengue virus (DENV) capsid degraders were designed using a known capsid inhibitor ST148 as a targeting ligand and cereblon and VHL-1 ligands. The authors identified two PROTAC DENV capsid degraders, namely RPG-01-132 and RPG-01-139 for further investigation. Their data showed that RPG-01-132 and RPG-01-139 induced degradation DENV capsid via a PROTAC mechanism. Furthermore, the authors further showed that RPG-01-132 inhibited multiple functions of the dengue capsid protein and demonstrated a broad activity against dengue virions, as compared to ST148. Overall, this is a good proof-of-concept study that induced degradation of the dengue virus capsid protein could be a potential therapeutic strategy for the treatment of the dengue virus infection.

Response: We thank Reviewer 2 for positive feedback and for the overall favorable assessment of our study.

1. Some revisions are recommended. First, RPG-01-132 (and also RPG-01-139) induced degradation of GSPT1 protein, which may contribute to the activity of RPG-01-132 on inhibition of multiple functions of the dengue capsid protein and its broad activity against dengue virions. In the study, the authors compared the potencies RPG-01-132 and RPG-01-139 vs two known and very potent GSPT1 inhibitors and concluded that because RPG-01-132 and RPG-01-139 only induced degradation of GSPT1 at low micromolar concentrations and were much weaker than those two known GSPT1 degraders, their GSPT1 degradation activity did not contribute to their activity on inhibition of multiple functions of the dengue capsid protein and its broad activity against dengue virions. However, RPG-01-132 was not a very potent degrader and inhibitor of the dengue virus capsid protein and also had a micromolar potency against dengue virions. Hence, it is very likely that its GSPT1 degradation activity of RPG-01-132 contributes to the degradation of the dengue capsid protein, as well as its activity against dengue virions.

Response: We thank Reviewer 2 for prompting us to consider this issue more carefully and to try to present our logic more clearly. RPG-01-132's DC₅₀ value against GSPT1 is 7.2 μM (**Supplementary Fig. 14d, e**). At this concentration, we observe robust antiviral activity with the viral yield reduced by ~80% (**Fig. 4a**). At a concentration of 40 nM CC-90009, corresponding to >10-fold more than its DC₅₀ (DC₅₀ 3.7 nM), the reduction in viral yield is less than 20% (**Supplementary Fig. 14c, g**). Based on this, we conclude that depletion of GSPT1 can contribute to antiviral activity, but it is not the predominant source of antiviral activity for RPG-01-132 since we would otherwise expect to see comparable (if not much higher) levels of antiviral activity for CC-90009 at 10x its DC₅₀ concentration. We agree with the Reviewer, however, that even a minor contribution of GSPT1 depletion to antiviral activity should be acknowledged, and have added the following text to the manuscript:

"Since GSPT1 inhibition is known to be antiviral for several DNA and RNA viruses¹⁰⁻¹², we examined if GSPT1 degradation contributes to the antiviral activity exerted by RPG-01-132 by comparing its GSPT1 DC₅₀ and antiviral EC₅₀ values with those of CC-90009¹³ and SJ6986¹⁴, two potent and selective GSPT1 degraders. CC-90009 and SJ6986 exhibited DC₅₀ values against GSPT1 in the nanomolar range in Huh7.5 cells (**Supplementary Fig. 14c**). In contrast, RPG-01-132 is a much weaker degrader of GSPT1, with DC₅₀ value of 7.2 μM (**Supplementary Fig. 14d, e**). At this concentration, RPG-01-132 reduces DENV2 yield by ~80% (**Fig. 4**). In contrast, at a concentration of 40 nM CC-90009, corresponding to >10-fold more than its GSPT1 DC₅₀ (DC₅₀ 3.7 nM), the reduction in viral yield is less than 20% (**Supplementary Fig. 14c, g**). Based on this, we conclude that although depletion of GSPT1 can have a very modest antiviral effect, it is not a major source of RPG-01-132's antiviral activity since we would otherwise expect to see comparable (if not much higher) levels of antiviral activity for CC-90009 at 10x its DC₅₀ concentration."

2. Second, as compared to many reported PROTAC degraders, the degradation potency (DC₅₀) and

efficiency (D_{max}) are very modest. While this study provided a proof-of-concept, the therapeutic application of a PROTAC degrader of the dengue virus capsid protein will be very limited. Please discuss this important limitation in the paper.

Response: The Reviewer makes a point that is well-taken in terms of the broader, long-term impact of this work. This point was also made by Reviewer 3 (comment 1). We think it's important to point out that essentially all heterobifunctional degraders that have made it to the clinic started with leads that required extensive optimization to improve potency and ADME/PK. For example, early degraders targeting the estrogen receptor had only micromolar DC_{50} values at the proof-of-concept stage¹⁵⁻¹⁹, but iterative structural optimization ultimately yielded clinical candidates ARV-471²⁰⁻²² and AC0682²³ whose DC_{50} values are, low to sub-nanomolar. Similarly, initial androgen receptor degraders with modest degradation potency^{15,19} were optimized through extensive medicinal chemistry to generate clinical candidates including ARV-110^{24,25} and CC-94676²⁶. From these and many other examples, we think the TPD field accepts that this type of optimization is possible. The larger question in our minds, and we think for the field, is what antiviral targets would justify this type of effort? Due to the limited number of validated antiviral degraders in the published literature and the limited evidence that highly abundant viral structural proteins are suitable targets for an antiviral TPD strategy, we thought it important to focus our efforts in this study on proof-of-concept that (i) we can achieve significant degradation of capsid protein, (ii) that this depletion of capsid results in significant antiviral activity, and (iii) characterizing the differentiated pharmacology that results from targeted degradation rather than inhibition of capsid, including resilience to ST-148-resistance and impact(s) on nonstructural functions of capsid. We agree wholeheartedly that RPG-01-132 is a tool compound and additional chemistry is needed to generate a preclinical lead with appropriate potency and drug-like properties for further development, and have added the following text to the Discussion to highlight this:

"Demonstration of antiviral efficacy *in vivo* is a necessary next step in the preclinical development of TPD-based antivirals, This will require improvements in potency, cell permeability, and other ADME-PK parameters²⁷ with iterative rounds of medicinal chemistry to optimize linkers and E3 ligands, and may additionally require the identification of improved ligands for the DENV C protein. While success in this endeavor cannot be assumed, optimization of heterobifunctional degraders targeting BRD4, BCL6, estrogen receptor, androgen receptor, and many other targets to clinical stage candidates demonstrate that this preclinical and clinical optimization is possible. This study validates DENV C as a target for TPD DAA development and establishes the tools needed for this goal."

Reviewer #3 (Remarks to the Author):

In this study, Chakravarty and colleagues report a novel class of dengue virus inhibitors which seemingly induce the degradation of the capsid viral protein (C). Following screening and characterization, the authors focus the study on their best compound, RPG-01-132, which possesses an antiviral potency in the micromolar range for dengue virus serotype 2 based on the determination of the EC_{50} value. Notably using KO cells and control compounds, the authors convincingly demonstrate that the decrease in C expression in DENV-infected cells is dependent on CRBN.

Response: We thank the Reviewer for the supportive comments regarding our convincing demonstration of targeted degradation of C and the elegance of this approach.

1. Although, this approach is elegant and make the proof-of-concept that it can identify novel classes of antivirals in principle, the antiviral potency of the drug candidates was rather low. Indeed, the EC_{50} of RPG-01-132 was around 1 μ M against DENV serotype 2 while the maximum reduction of viral titers in treated infected cells was less than 10-fold. This was in fact lower than the phenotype observed with ST148, the (non-degrader) capsid inhibitor whose backbone was used to generate the studied

compounds. From a drug development point of view, such activity is not sufficient, especially since the lead compound has a very poor antiviral activity against dengue viruses from serotypes 1,3, and 4 while it was not challenged in in vivo infection models.

Response: This comment partially overlaps with Reviewer 2's, Comment 2 so we here reference our response to that comment. To summarize, optimization of RPG-01-132's potency, selectivity, and ADME/PK properties will be necessary to generate a preclinical candidate. This optimization is a necessary next step in developing DENV capsid degraders as DAAs and may also include the identification of new C-targeting ligands. Although success in the significant effort that this optimization will require is not guaranteed, the TPD field has had collective success in optimizing other heterobifunctional degraders from initial proof-of-concept molecules with DC₅₀ values in the low micromolar to clinical stage degraders with DC₅₀ values in the low- to sub-nanomolar. We therefore believe that this will also be possible for the DENV C degraders. In our view, the field needed us to answer several critical questions to establish that this kind of optimization effort would be worth launching, including:

- (i) Can the capsid be targeted for degradation? It might not be a substrate compatible with the E3 ligases for which we have ligands; moreover, its oligomerization and localization might protect it from the ubiquitin-proteasome machinery.
- (ii) Even if the capsid can be degraded via a TPD strategy, can one deplete enough capsid to have significant antiviral activity? Given its function as a structural protein, the capsid is present in higher relative abundance than viral enzymes, and the threshold amount of capsid needed to allow production of infectious virions is unknown. Conversely, the extent of capsid depletion required for antiviral activity was not known but speculated to be quite high.
- (iii) Can a TPD strategy be used to target functions of viral proteins that have been difficult to inhibit and exert antiviral activity by inducing the loss of multiple functions of viral proteins? This is important for the field, given the number of "undruggable" viral proteins that have been recalcitrant to both phenotypic and target-based inhibitor strategies and the prevalence of multifunctional viral proteins.

We believe that our study is valuable to the antivirals and TPD fields in providing decisive, affirmative answers to these questions and by laying a foundation for preclinical development of DENV capsid degraders.

The Reviewer also rightly pointed out that RPG-01-132's antiviral potency is actually lower than that of ST148. This has often been initially observed by the TPD when inhibitors are converted to heterobifunctional degraders due to reductions in cell penetrance, and our pilot studies indicate that this is the case for RPG-01-132 compared to ST148 (**Supplementary Table 1**). Since ultimately antiviral potency is what matters for drug development, we have not tried to normalize for the difference in cell penetrance, and we do not think that the potency of RPG-01-132 undermines any of our conclusions. We also think it is likely that poor cell penetrance and limited solubility prevent us from dosing RPG-01-132 at concentrations that inhibit DENV3 and DENV4; however, compounds with improved capsid engagement and cell penetrance are needed to demonstrate this explicitly. We believe that the iterative design-make-test cycles needed to improve cell penetrance, potency, selectivity, and ADME/PK, while definitely worthwhile efforts, go beyond the scope of the current work. Likewise, demonstration of activity in an animal model would require work that goes well beyond the current study since many CRBN-based degraders lack activity in murine models due to species differences in human versus mouse CRBN and since optimization of the capsid-targeting moiety to improve the suboptimal ADME/PK of ST148 will likely be a necessary part of the next phase of capsid degrader development.

2. It is interesting that it is still active against an ST148-resistant DENV2 strain; but this does not preclude

that the virus will adapt rapidly to RPG-01-132, even with compensatory mutations simply increasing replication.

Response: This critique is related to Comment 4 made by Reviewer 1, so we refer Reviewers to our response to that comment as well. We agree with the Reviewers that we cannot exclude the possibility that DENV2 will adapt to RPG-01-132 and become less sensitive due to mutations that decrease binding and/or increase replication. We have not performed serial passage of DENV2 in the presence of RPG-01-132, based on the higher biosecurity constraints placed on this type of experiment, which our institutional biosafety committee has informed us is now classified by NIH as gain-of-function research, and based on the open-endedness of this type of experiment. As an example, in our work with HCV NS3 degraders, conducted before the current restrictions on gain-of-function research, we did not observe the emergence of a quasi-species resistant to the NS3 degrader after > 10 passages. This was a negative result and still did not prove that resistance cannot evolve. To ensure our ability to draw a conclusion, we did the head-to-head comparison of ST148 and RPG-01-132 with the only mutation known to confer resistance to ST148 and have been careful in our manuscript to indicate that our results with the DENV2 C-S34L mutant support the idea of better resilience to point mutations that affect drug-binding. Since other antivirals with TPD mechanisms have not reported resistance studies, either through passaging or site-directed mutagenesis, we respectfully point out that our characterization exceeds the current standard in the field. This stated, we believe that we stand in agreement with both Reviewers 1 and 3 that issues of resistance need to be examined across many more targets and small molecules for the field to be able to generalize conclusions regarding targeted protein degradation mechanisms and barriers to resistance.

3. The authors claim that the activity of the RPG-01-132 is capsid-specific. It is indeed quite clear that in drug-treated infected cells, C levels decrease in a drug-concentration- and CRBN-dependent manner. However, in infected cells, the expression of C as well as that of all viral proteins is directly dependent on the abundance of viral RNA, and thus on the efficiency of RNA replication. Considering this, if the drug targets the RNA replication step of the life cycle (instead of C as concluded by the authors), it is expected that this will result in the decrease in viral protein expression independently of C. Thus, to prove that the drug mode-of-action is C-dependent/specific, the authors must evaluate the impact of drug treatment on: 1- the expression of other structural and nonstructural viral proteins using western blotting (which should not change if the hypothesis is correct), 2- the levels of viral RNA (to show that replication is not inhibited), and 3- the levels of untagged C, when expressed alone in a replication-independent context. Measuring C expression in a luciferase assay using a tagged protein is not appropriate, especially since the data of Fig S10 are misleading. With this indirect readout, protein expression seems to increase upon drug treatment, which is the opposite of what is expected.

Response: 1- We thank Reviewer 3 for prompting us to provide additional data demonstrating that the antiviral activity is capsid-specific. As Reviewer 1 had a related critique in Comment 2, we also refer Reviewer 3 to our response there. We conducted the three experiments requested by Reviewer 3 as follows:

- (1) We conducted Western blots to show that RPG-01-132 does not affect the abundance of DENV2 NS4B, NS5, or envelope proteins (**Supplementary Fig. 11**).
- (2) We verified by RT-qPCR that RPG-01-132 does not affect the level of viral RNA (**Supplementary Fig. 12b**) and further used the subgenomic replicon system to show that viral RNA replication is not affected by the capsid degrader (**Supplementary Fig. 13**).
- (3) We expressed untagged DENV2 capsid in HEK293 cells and monitored capsid abundance by Western blot after treatment with RPG-01-132 for 24 h (**Supplementary Fig. 17g**). The data reveal a biphasic profile that parallels the concentration-dependent nLuc activity generated by the C-nLuc reported in **Supplementary Fig. 17a**. That is, we observe reduced untagged capsid at

concentrations at which we observed reduced nLuc activity, consistent with comparable targeted degradation of untagged capsid and capsid-nLuc. In contrast, at concentrations > 0.62 μM , we observe a relative increase in untagged capsid abundance, which also parallels the increase in nLuc signal observed in the capsid-nLuc experiments. Increased untagged capsid and capsid-nLuc at higher concentrations of RPG-01-132, which is consistent across these experiments, may be due to a hook effect that disfavors ternary complex formation (see also response to Reviewer 1, Comment 3) and/or stabilization of capsid, as has been reported for ST148. All data from these Supplementary Figures have been reproduced below.

The following text is added to the “Results” section: “Since changes in nLuc activity might not reflect C-nLuc abundance in this system, we verified that this concentration range of RPG-01-132 causes depletion of untagged DENV2 capsid ectopically expressed in HEK293 (Supplementary Fig. 17g).”

Supplementary Figure 11. Effect of RPG-01-132 on other viral proteins in DENV2-infected cells. (a–b) Western blots of NS4B from DENV2 NGC–infected cells (MOI 0.5) treated with a concentration series of RPG-01-132 (20 μM –40 nM) or RPG-01-132-BUMP. Lysates are the same samples analyzed in Fig. 3b. The abundance of NS4B is unchanged by all compounds relative to the DMSO control. (c) Western blot analysis of E and NS5 from parallel infections (MOI 0.5) treated with 1 μM ST148 or 2.5 μM RPG-01-132 or RPG-01-132-BUMP, corresponding to concentrations ~5-fold higher than their respective antiviral EC_{50} values determined in Fig. 4 and Supplementary Fig. 6. The abundances of E and NS5 are comparable to those observed for the DMSO control. Representative blots are from $n = 2$ independent experiments.

a

b

Supplementary Figure 12. Determination of RPG-01-132's effect on steady-state abundance of viral RNA. (a) Schematic of the experiment (created in BioRender). Huh7.5 cells infected with DENV2 NGC strain (at MOI 0.5) were treated with RPG-01-132 (2.5 μ M), its negative control RPG-01-132-BUMP (2.5 μ M), or parental inhibitor ST148 (1 μ M) at concentrations corresponding to five-fold greater than the antiviral EC₅₀ values for the respective compounds as established in **Fig. 4** and **Supplementary Fig. 6**. Following incubation for 2 h and 8 h, cells were harvested, and total RNA extracted for cDNA synthesis and subsequent qPCR using primers specific for the capsid gene. (b) RT-qPCR analysis of dengue virus capsid RNA abundance. No significant changes were observed in RPG-01-132 compared to the DMSO control, indicating that reductions in capsid protein and viral yield are not due to effects of the compounds on viral RNA. Representative data are from $n = 2$ independent experiments.

a**293-T-Rex DVGZ (293-T-Rex cells expressing DENV2 sub-genomic replicon)****b****c****d****e****f****g****h****Supplementary Figure 13. Effect of RPG-01-132 in the DENV2 subgenomic replicon model.**

(a) Schematic of the subgenomic DENV2 replicon stably harbored by 293-T-Rex-DVGZ cells, a well-validated model for studying DENV translation, transcription, and RNA replication⁵. Accumulation of viral genomic RNA can be reliably monitored by detection of the eGFP reporter encoded on the replicon RNA.

(b-f) 293-T-Rex-DVGZ cells were treated with RPG-01-132, RPG-01-132-BUMP (20 μM-20 nM), ST148, or the NS4B inhibitor NITD-688 (20 μM - 10 nM). Replicon activity was measured by GFP fluorescence from 0-40 h after compound addition. NITD688 potently inhibited replicon replication, validating the assay. In contrast, RPG-01-132, RPG-01-132-BUMP, and ST148 had no effect in this system, consistent with a capsid-specific mechanism rather than a more general effect on viral RNA translation, transcription, or replication. **(g,h)** Western blot for NS4B to verify that the reduced eGFP reporter signal reflects inhibition of replication and a decrease in viral protein following 24-hour treatment with NITD688 and no change in NS4B abundance caused by RPG-01-132.

Supplementary Figure 17. DENV C degrader RPG-01-132 affects the nonstructural functions of C (related to Fig. 5). (a) DENV C-NanoLuciferase (C-nLuc) fusion protein was expressed in HEK293 cells and treated with RPG-01-132 at concentrations 10 nM to 10 μM . RPG-01-132 shows a consistent $\sim 50\%$ depletion of C-nLuc at 0.625 - 0.312 μM . Representative data are shown from $n = 4$ independent replicates. (b-f) Schematic of the experiment is shown in Fig. 5. HEK293 cells were first transfected with a firefly luciferase (FF-Luc)-expressing plasmid under the control of an IFN- β promoter. 24 hours later, these cells were transfected with the C-nLuc plasmid and treated with 0.625, 0.312, and 0.156 μM concentrations of ST148, RPG-01-132, RPG-01-132-BUMP, or DMSO as negative control. At this step, we also co-transfected the cells with high molecular weight poly(I:C) to activate the IFN- β promoter.

Twenty-four hours later, IFN- β promoter activity was detected by measurement of FF-Luc activity and C-nLuc abundance was detected by measurement of nLuc activity. **(b-c)** Transient expression of DENV C suppresses IFN- β promoter activation. RPG-01-132 rescues the promoter activity in a concentration-dependent manner, whereas ST148 and RPG-01-132-BUMP do not. Data from one experiment are shown in **Fig. 5c**, and data from the other two independent experiments are shown here (replicates 2 and 3). **(d-f)** After the detection of FF-Luc signal, nLuc signal was detected. Representative data are shown for replicates 1, 2, and 3. **(g)** To show that the reduction of nLuc activity caused by RPG-01-132 is due to degradation of C protein and not an artifact of the nLuc tag, an untagged DENV2 C was expressed in HEK293 cells and treated with a dilution series of RPG-01-132 (10 μ M- 2 nM) or DMSO for 24h. Lysates were analyzed by Western blot. Quantification beneath lanes shows C signal normalized to GAPDH and expressed relative to DMSO (=100). Higher concentrations of RPG-01-132 (10–2.5 μ M) appear to stabilize C, consistent with the activity of parental inhibitor ST148; however, C abundance is decreased at lower concentrations, with a minimum (~50% of DMSO) at 0.62 μ M and returning to near baseline at concentrations \leq 0.31 μ M. This parallels the effect of RPG-01-132 on nLuc activity when C-nLuc is ectopically expressed. The red box in **a** and **g** indicates the same concentration range.

4. Even if the two drugs are structurally related, the mode-of-action of ST148 and RPG-01-132 are different. Given that, as explained above, the C-specificity of RPG-01-132 is not addressed, the authors must confirm that this drug is targeting the C-dependent steps of the life cycles, namely viral assembly and entry/uncoating. They should test the drug on several assays such as time-of-addition assays (to test entry), compare RNA levels and the production of infectious particles in infected cells to ensure that the decrease in titers is not due to a reduction of replication, and confirm that the drug does not impact the replication of a DENV sub-genomic replicon (which does not express C and does not make viral particles).

Response: We thank Reviewers 3 and 1 for prompting us to provide additional data demonstrating capsid-specific antiviral activity and mode of action. As outlined in our response to Reviewer 1, Comment 2, and in part in response to Reviewer 3's Comment 3 above, we verified that RPG-01-132

(1) does not affect the abundance of viral RNA (**Supplementary Fig. 12a, b**)

(2) does not exhibit activity in a subgenomic replicon system (**Supplementary Fig. 13**) that reports on viral translation and genome replication but lacks steps of viral entry/uncoating, and assembly

(3) does not impact the abundance of other DENV proteins (envelope, NS4B, NS5) (**Supplementary Fig. 11**).

Although our current experiments do not allow us to discern if antiviral activity due to capsid depletion results from effects early or late (or both) in the replication cycle, our data collectively address the Reviewers' request that we show that RPG-01-132 acts at capsid-specific steps of the replication cycle.

All three of these figures were included in our response to the Comment immediately preceding this one.

The following text has been added to the "Results" section:

"Since the degrader antiviral activity is notably observed at concentrations that cause less than 50% depletion of intracellular C, this could suggest that even a modest depletion of C is sufficient to block the assembly of DENV virions. Alternatively, this might indicate that RPG-01-132 and RPG-01-139 exert antiviral activity through mechanisms other than loss of capsid function. For example, any mechanism inhibiting replication of the viral RNA genome would also be expected to cause depletion of not only C but all viral proteins over time, since the template for translation of these proteins would be reduced. To test our hypothesis that RPG-01-132's antiviral activity is due to specific loss of C, we conducted additional experiments to rule out more general mechanisms. First, we reasoned that if RPG-01-132's

antiviral activity is due to a more general effect on replication, this would be reflected in depletion of other DENV proteins and not only C. Experiments examining the effect of RPG-01-132 on the abundance of envelope (E), NS4B, and NS5 detected no changes (**Supplementary Fig. 11**), confirming that the depletion is specific for C and disfavoring the idea that RPG-01-132 has a general effect on viral replication or viral RNA. As another, more explicit test of whether RPG-01-132's antiviral activity is due to more general effects on viral replication, we examined whether RPG-01-132 changes the steady-state abundance of the viral genomic RNA and found no effect on C RNA abundance (**Supplementary Fig. 12**). We also corroborated that RPG-01-132 does not affect translation or replication of the viral RNA using a DENV2 subgenomic replicon system⁵, which lacks the viral structural genes, including C, and is a well-established model for studying DENV RNA translation and replication in the absence of the steps of viral entry and virion assembly. Whereas the NS4B inhibitor NITD688 caused a reduction in replicon activity, RPG-01-132 and ST148 had no effect in this model (**Supplementary Fig. 13**), consistent with the idea that their antiviral activities are capsid-specific. Together, these experiments rule out the possibility that RPG-01-132's antiviral activity is due to more general effects on DENV replication and support the hypothesis that its effect on DENV is due to specific loss of C protein."

5. All western blots need to be quantified to help the reader to evaluate the phenotypes. This is especially true for Fig 2 (the starting point of the study) and Fig S3, which as such is not convincing. Indeed, in this figure, changes in C and CRBN stability is really not obvious since the reference protein levels (100%) at the lowest temperature (48°C) change among drug treatments.

Response: We thank Reviewer 3 for prompting us to address this. The Western blot assays we conducted can only be regarded as semi-quantitative and suitable for comparisons between samples analyzed on the same blot because we did not load known quantities of capsid protein standards and because there are limitations in signal linearity, exposure dependence, and normalization variability between different blots. In Fig. 2, at the outset of this study, we were using the Western blot assay primarily as a rapid 'thumbs-up/thumbs-down' screen to triage compounds and so did not try to quantitate. At Reviewer 3's request, we performed densitometry on the Western blots shown in **Fig. 2** and **Fig. 3b** and provide these data now in the new **Supplementary Figs. 2a-e** and **7**. They are also included in our response to this comment below

The following sentence has been added to the manuscript as part of the legend for **Fig. 2**:
"Densitometric quantification of the blots in **Fig. 2** is provided in **Supplementary Fig. 2a-e**."

Reviewer 3 also had well-taken concerns regarding **Supplementary Fig. 4 (previously Supplementary Fig. 3)** due to our having analyzed compound-treated samples and DMSO control samples on different blots, which complicated normalization across temperatures. To address this concern and provide additional data supporting intracellular target engagement, we performed a single-temperature CETSA in infected Huh7.5 cells to demonstrate compound-dependent stabilization of capsid by ST148, RPG-01-132, and RPG-01-139 and selective stabilization of CRBN by CRBN binders (**Supplementary Fig. 4c**). These new data are reproduced below.

Supplementary Figure 7. Densitometric quantification of blots from Fig. 3b. Western blot signals for C were quantified by densitometry, normalized to GAPDH loading control, and expressed relative to DMSO (=100). Data correspond to the blots shown in Fig. 3b and confirm the concentration- and CRBN-dependent decrease in C in the presence of RPG-01-139 and RPG-01-132.

C

Supplementary Figure 4c. To avoid variability in CETSA samples analyzed on different blots, we performed a single-temperature assay in infected Huh7.5 cells, with all samples analyzed in parallel on the same gel to enable comparison of relative amounts of C and CRBN and detection of evidence of stabilization of these proteins relative to the DMSO-treated controls. Huh7.5 cells (DENV2 NGC MOI = 1) were treated for 2 h with DMSO, ST148, lenalidomide, RPG-01-139, RPG-01-139-BUMP, RPG-01-132, or RPG-01-132-BUMP at 10 μ M. Lysates were subjected to thermal challenge at 50.1 °C (C) or 55.2 °C (CRBN) and analyzed by Western blotting on a single gel per target together with detection of the GAPDH loading control. Band intensities were quantified by densitometry, normalized to GAPDH, and expressed relative to DMSO. Representative values under each lane are presented. Data show selective stabilization of C by ST148, the C degraders, and their respective BUMP controls, and specific stabilization of CRBN by lenalidomide and the C degraders. Schematics in (a, b, and c) were created in BioRender.com.

6. Figure 5 is not convincing at all with the provided image quality. As such, it is hard for the reader to evaluate that the arrows indicate virion as stated. They do not look like the previously reported regular arrays. Magnified images are required to appreciate this. Moreover, the indicated viral factories (VF) do not resemble the endoplasmic reticulum-derived invaginated vesicles typically induced by DENV. One could argue that these structures are instead reminiscent of autophagosomes and not VFs. More importantly, why VFs (which host RNA replication) would be impacted by the drug treatment if the compound targets viral assembly. Alternatively, if the drug targets replication, it is expected that less VFs will be observed, simply because there are less viral RNA and viral proteins, and not because VF biogenesis is directly targeted by the drug. This needs to be rationalized.

Response: We thank Reviewer 3 for these constructive comments, which are also related to those in Comment 10 (below). First, we apologize for incorrect usage of the term "viral factory," which should refer

to the sites of viral RNA replication, not the sites of viral assembly. At the time point that we imaged, the predominant structures visualized are expected to be assembled virions that have already budded through the ER membrane and are trafficking within endoplasmic reticulum (ER) cisternae and Golgi. They are not sites of viral RNA replication. Second, we should have pointed out differences in our sample preparation compared to that of Scaturro *et al.* (J. Virol. 2014, Bartenschlager group). There, the authors²⁸ infected cells at lower MOI, fixed and then sectioned them directly on coverslips, whereas we harvested infected cells, fixed them as cell pellets that we then sectioned as blocks. Our protocol results in cells in different planes and, unfortunately reduces the clarity of virion arrays compared to those depicted in the Scaturro study. While the morphology in our experiments appears less regular, the presence of virion-like particles within ER cisternae and Golgi vesicles is evident. We remade **Fig. 5** with images that we think better show virion-like particles at the indicated sites and now provide additional representative raw images from each treatment in **Supplementary Fig. 15**. Our interpretation of the data is that RPG-01-132 reduces the abundance of capsid protein, consequently limiting the formation of assembled virions. This is consistent with the profound reduction in assembled virions visible in our samples, which were imaged at a relatively late stage of infection. We have corrected the text to reflect this interpretation and to avoid implying that RPG-01-132 directly impacts replication organelles. The updated text reads: "In contrast, the presence of RPG-01-132 results in the detection of very few viral particles in the ER or Golgi and the appearance of vesicular structures resembling autophagosomes and/or lysosomes (**Fig. 5a**). This suggests a block in viral assembly caused by the limited availability of C protein and is a phenotype that differs strikingly from that induced by ST148."

Figure 5: RPG-01-132 inhibits multiple functions of the dengue capsid protein. (a) Negative-stained thin sections were imaged by transmission electron microscopy to visualize the effects of inhibitor vs. degrader treatment on virion assembly. Clockwise from upper left, samples are mock-infected with DMSO vehicle, DENV2 NGC-infected with DMSO vehicle, DENV2 NGC-infected treated with RPG-01-132, and

DENV2 NGC-infected treated with ST148. Red arrows indicate the presence of DENV2 virions. M = mitochondria, V = virions. For each condition, two grids were screened, and three representative images are presented here and in **Supplementary Fig. 15**. The scale bar is 200 nm. **(b)** Schematic of the experimental design to investigate RPG-01-132's effect on the antagonism of the type I interferon response. In this dual luciferase system, a firefly luciferase reporter gene (FF-Luc) is regulated by the IFN- β promoter, and C is ectopically expressed as a fusion protein with nanoluciferase (C-nLuc) to allow monitoring of C abundance. ST148, RPG-01-132, and RPG-01-132-BUMP were used at 0.156, 0.312, or 0.625 μ M. Image created in BioRender.com. **(c)** Only RPG-01-132 rescues IFN- β promoter activity in a concentration-dependent manner; ST148 and RPG-01-132-BUMP do not, indicating that this effect is dependent on targeted degradation of DENV C. The IFN- β reporter assay measures luminescence produced by the RLU (Relative Light Units), the standard output of the luciferase assay. Representative data from $n = 3$ independent experiments are shown, with the other experiments presented in **Supplementary Figs. 17b-c**. Data for nLuc-based detection of C-nLuc abundance are provided in **Supplementary Fig. 17d-f**.

Additional comments:

7. - What is the impact of the treatment of all the drugs of Fig 1 on viral titers. This would be more informative for an antiviral potency than C expression

Response: With all respect to Reviewer 3, at this stage of the study, our primary goal was to identify heterobivalent compounds that have changed mechanism from ST148's stabilization of capsid dimers to a TPD mechanism. Since the parental compound has antiviral activity and reduced viral titers, measuring viral yield may not be helpful in discerning which compounds have TPD activity. This led us to focus on Western blots for capsid abundance. In terms of our manuscript providing a "road map" for others interested in converting antiviral inhibitors to antiviral degraders, we believe that monitoring target abundance may in general be a more expedient and informative assay by which to advance candidate molecules for further improvement and study as TPD agents. This stated, out of deference to Reviewer 3, we have performed the requested viral titer experiments and now include the data in the **Supplementary Fig. 2f-j**.

We have added the following text in the Results section:

"While we also evaluated effects of candidate degraders on viral titers (**Supplementary Fig. 2f-j**), we ultimately found monitoring C abundance to be the more efficient assay for distinguishing TPD activity from the activity of parental ligand ST148."

Supplementary Figure 2. Densitometric quantification of Western blots from Fig. 2 and antiviral activity. (a-e) Densitometric quantification of DENV C Western blots. For each sample, the C signal was normalized to the signal for the GAPDH loading control and is expressed here as a percentage of the DMSO control, which was set at 1. Data correspond to the blots shown in Fig. 2. **(f-j)** Effect of candidate C degraders on the titer of infectious DENV2 secreted to culture supernatants. Viral titers were quantified by viral plaque formation assay. Co-treatment with lenalidomide or VH032 together with vehicle (DMSO) did not significantly alter DENV titers. ST148 consistently inhibited viral replication via a mechanism that was not affected by the presence of the E3 ligand. RPG-01-139 and RPG-01-132 exhibited antiviral activity that was rescued in the presence of lenalidomide, mirroring the depletion of C and rescue in the

Western blot assay. Compounds LNW-01-108, LNW-01-100, and LNW-01-148 exhibited antiviral activity without detectable depletion of C, indicating a non-TPD mechanism. VHL-based compounds also lacked TPD-based antiviral activity based on the lack of rescue in the presence of excess VHL ligand as a competitor.

8.- Fig S4: Although this is inferred, what FKBP12 western blot control for. This is not indicated.

Response: We apologize for forgetting to explain this and thank Reviewer 3 for bringing this to our attention. This was a control to validate our knockout cell line by showing that the validated, commercially available FKBP12 degrader dFKBP1 loses its activity in our CRBN KO cells. We have added the following text to the legend of **Supplementary Fig. 4** and to the Methods section describing generation of the CRBN knockout cells:

“We validated functional knockout of CRBN by showing that dFKBP-1²⁹, a validated CRL4^{CRBN}-dependent degrader of FKBP12, has concentration-dependent degrader activity against FKBP12 in wild-type Huh7.5 cells but not in Huh7.5-CRBN^{-/-} cells (**Supplementary Fig. 5**).”

9.- Why was the proteomic study performed in MOLT4 cells while the rest of the study was done exclusively in Huh7.5 cells?

Response: The purpose of the proteomics assessment was to profile substrates targeted for degradation by RPG-01-132. Since this degrader uses a CRBN ligand, we were particularly concerned with detecting any activity against Ikaros transcription factors and other known targets of CUL4^{CRBN}. MOLT4 cells are one of the most established and widely used models for proteomic profiling of CRBN-based degraders due to the high abundance of IKZF proteins in these cells³⁰⁻³³. These CRBN targets are not present at high enough abundance in Huh7.5 cells to permit reliable detection of any degradation activity against these "off-targets." We have added the following text to the Methods section:

"MOLT4 cells were utilized for these studies to enable sensitive detection of degradation of the IKZF proteins targeted by CRBN ligands, as detection of activity against these potential "off-targets" is challenging in Huh7.5 due to low expression."

10.- It is unclear why the authors state that the increase in lysosomes abundance is indicative of an aborted replication process. It could simply reflect that the autophagy is induced, which is not per se an antiviral process.

Response: Reviewer 3 makes a good point about this here and in Comment 6. We agree that (1) increased lysosomes alone cannot be taken as definitive evidence of aborted replication; (2) autophagy can be induced by many stressors and is not necessarily antiviral; and (3) we cannot definitively classify the vesicular structures we observe in the RPG-01-132-treated samples as lysosomes or autophagosomes since we did not perform immunogold or other labeling for lysosomal or autophagosomal markers. In our study, RPG-01-132 depleted intracellular C. This was accompanied by a reduction in the appearance of virion-containing ER cisternae (**Fig. 5**) and an increased appearance of vesicles that could be lysosomes and/or autophagosomes. We have revised the original language to more correctly reflect this interpretation and to avoid overstatement as follows:

"In contrast, the presence of RPG-01-132 results in the detection of very few viral particles in the ER or Golgi and the appearance of vesicular structures resembling autophagosomes and/or lysosomes (Fig. 5a)."

11.- Fig S6: It would be helpful that the BRET assay principle is explained in more details. A schematic could be relevant.

Response: A schematic of the nanoBRET assay has been added as **Supplementary Fig. 8a**.

Supplementary Figure 8. Detection of ternary complex formed by RPG-01-132 with CRBN and DENV capsid in live cells. (a) Schematic of NanoBRET® assay used to detect induced proximity of CRBN or ubiquitin with DENV capsid protein. Created in BioRender.com.

12.- What is the impact SJ6986 treatment on DENV titers?

Response: SJ698 causes a very modest decrease in DENV2 titer. At a concentration of 39 nM, which is almost 4 times the DC₅₀ for GSPT1, DENV titer is reduced by less than two-fold. The viral titer data for SJ698 have been added as **Supplementary Fig. 14h**. As described in our response to Reviewer 2's Comment 1, we have modified the text to read as follows:

"Since GSPT1 inhibition is known to be antiviral for several DNA and RNA viruses¹⁰⁻¹², we examined if GSPT1 degradation contributes to the antiviral activity exerted by RPG-01-132 by comparing its GSPT1 DC₅₀ and antiviral EC₅₀ values with those of CC-90009¹³ and SJ6986¹⁴, two potent and selective GSPT1 degraders. CC-90009 and SJ6986 exhibited DC₅₀ against GSPT1 in the nanomolar range in Huh7.5 cells (**Supplementary Fig. 14c**). In contrast, RPG-01-132 is a much weaker degrader of GSPT1, with DC₅₀ value of 7.2 μM (**Supplementary Fig. 14d, e**). At this concentration, RPG-01-132 reduces by ~80% (**Fig. 4**). In contrast, at a concentration of 40 nM CC-90009, corresponding to >10-fold more than its GSPT1 DC₅₀ (DC₅₀ 3.7 nM, empirical estimate), the reduction in viral yield is less than 20% (**Supplementary Fig. 14c, g**). Based on this, we conclude that although depletion of GSPT1 can have a very modest antiviral effect, it is not a major source of RPG-01-132's antiviral activity since we would otherwise expect to see comparable (if not much higher) levels of antiviral activity for CC-90009 at 10x its DC₅₀ concentration."

Supplementary Figure 14. Analysis of DENV C degrader selectivity. The substrate repertoires of (a) RPG-01-139 and (b) RPG-01-132 were characterized in mass spectrometry-based proteomics experiments. MOLT4 cells were used to enable detection of IMID-based molecular glue degrader activity against IKZF1/3, which are not robustly expressed in Huh7.5. MOLT4 cells were incubated with the respective degraders at a concentration of 5 μM for 8 hours. GSPT1 is the main off-target of RPG-01-132. (c, d) Western blot analysis of GSPT1 protein in uninfected Huh7.5 cells treated with (c) GSPT1-specific degraders CC-90009 or SJ6986 or (d) C degraders RPG-01-139 and RPG-01-132. Nonlinear regression yielded DC_{50} values of 3.7 nM (CC-90009) and 13 nM (SJ6986). Because the curves did not fully saturate at the low-concentration plateau, these values are extrapolated and should be interpreted as approximate upper-bound estimates. Lanes with $\sim 50\%$ GSPT1 depletion are indicated with "*" in (c). (e) Determination of RPG-01-132's DC_{50} against GSPT1 by nonlinear regression analysis of data obtained by densitometric quantification of Western blot in (d). (f) DENV C protein in cell lysates examined by Western blot and (g-h) viral yield in culture supernatants measured by viral plaque formation assay following 24 hours of incubation of DENV2-infected (MOI 0.5) Huh7.5 cells with CC-90009 and SJ6986.

13.- Fig 5C: in the Y-axis, it should be FLU and not RLU

Response: We thank Reviewer 3 for this comment, as it showed us that we needed to make our description clearer to avoid confusing readers. **Figure 5c** reports data from a firefly luciferase (FF-Luc)-based reporter assay in which the FF-Luc gene is under control of the IFN- β promoter. The y-axis is correctly labeled in RLU (Relative Light Units), which is the standard readout of luminescence intensity generated by the FF-Luc reporter. We have clarified this in the figure legend to avoid confusion.

We have added the following clarification to the figure legend: “The IFN- β reporter assay measures luminescence produced by the RLU (Relative Light Units), the standard output of the luciferase assay.”

14.- What is the impact of ST148 treatment in the IFNbeta induction assay?

Response: We did not observe an effect of ST148 on polyIC-induced activation of the IFN- β promoter in our experiments. This could reflect a lack of effect of ST148 on this function of capsid protein and/or may indicate that ST148 was not present at a concentration that affects this function in the ectopic overexpression system. We have added the following text in the results section:

“Consistent with the idea that RPG-01-132's rescue of the IFN response in this system is due to targeted degradation of C, neither ST148 nor RPG-01-132-BUMP restored activation of the IFN promoter in the presence of C (**Fig. 5c, Supplementary Fig. 17b, c**). This suggests that degradation of C can block its antagonism of the IFN response, whereas ST148's stabilization of C dimers cannot, at least not in this ectopic expression system.”

References cited in point-by-point response

- 1 Riching, K. M. *et al.* Quantitative Live-Cell Kinetic Degradation and Mechanistic Profiling of PROTAC Mode of Action. *ACS Chem Biol* **13**, 2758-2770 (2018). <https://doi.org/10.1021/acscchembio.8b00692>
- 2 Mahan, S. D., Riching, K. M., Urh, M. & Daniels, D. L. Kinetic Detection of E3:PROTAC:Target Ternary Complexes Using NanoBRET Technology in Live Cells. *Methods Mol Biol* **2365**, 151-171 (2021). https://doi.org/10.1007/978-1-0716-1665-9_8
- 3 Park, D., Izaguirre, J., Coffey, R. & Xu, H. Modeling the Effect of Cooperativity in Ternary Complex Formation and Targeted Protein Degradation Mediated by Heterobifunctional Degraders. *ACS Bio Med Chem Au* **3**, 74-86 (2023). <https://doi.org/10.1021/acsbiochemau.2c00037>
- 4 Schwalm, M. P. *et al.* Tracking the PROTAC degradation pathway in living cells highlights the importance of ternary complex measurement for PROTAC optimization. *Cell Chem Biol* **30**, 753-765 e758 (2023). <https://doi.org/10.1016/j.chembiol.2023.06.002>
- 5 Ansarah-Sobrinho, C., Nelson, S., Jost, C. A., Whitehead, S. S. & Pierson, T. C. Temperature-dependent production of pseudoinfectious dengue reporter virus particles by complementation. *Virology* **381**, 67-74 (2008). <https://doi.org/10.1016/j.virol.2008.08.021>
- 6 Wurz, R. P. *et al.* Affinity and cooperativity modulate ternary complex formation to drive targeted protein degradation. *Nat Commun* **14**, 4177 (2023). <https://doi.org/10.1038/s41467-023-39904-5>
- 7 Riching, K. M., Caine, E. A., Urh, M. & Daniels, D. L. The importance of cellular degradation kinetics for understanding mechanisms in targeted protein degradation. *Chem Soc Rev* **51**, 6210-6221 (2022). <https://doi.org/10.1039/d2cs00339b>
- 8 Cecchini, C., Pannilunghi, S., Tardy, S. & Scapozza, L. From Conception to Development: Investigating PROTACs Features for Improved Cell Permeability and Successful Protein Degradation. *Front Chem* **9**, 672267 (2021). <https://doi.org/10.3389/fchem.2021.672267>

- 9 Haid, R. T. U. & Reichel, A. PK/PD modeling of targeted protein degraders: Charting new waters and navigating the shallows. *Drug Discov Today* **30**, 104311 (2025). <https://doi.org/10.1016/j.drudis.2025.104311>
- 10 Fang, J. *et al.* Functional interactomes of the Ebola virus polymerase identified by proximity proteomics in the context of viral replication. *Cell Rep* **38**, 110544 (2022). <https://doi.org/10.1016/j.celrep.2022.110544>
- 11 Fang, J. *et al.* Proximity interactome analysis of Lassa polymerase reveals eRF3a/GSPT1 as a druggable target for host-directed antivirals. *Proc Natl Acad Sci U S A* **119**, e2201208119 (2022). <https://doi.org/10.1073/pnas.2201208119>
- 12 Zhao, N. *et al.* Generation of host-directed and virus-specific antivirals using targeted protein degradation promoted by small molecules and viral RNA mimics. *Cell Host Microbe* **31**, 1154-1169 e1110 (2023). <https://doi.org/10.1016/j.chom.2023.05.030>
- 13 Surka, C. *et al.* CC-90009, a novel cereblon E3 ligase modulator, targets acute myeloid leukemia blasts and leukemia stem cells. *Blood* **137**, 661-677 (2021). <https://doi.org/10.1182/blood.2020008676>
- 14 Nishiguchi, G. *et al.* Identification of Potent, Selective, and Orally Bioavailable Small-Molecule GSPT1/2 Degraders from a Focused Library of Cereblon Modulators. *J Med Chem* **64**, 7296-7311 (2021). <https://doi.org/10.1021/acs.jmedchem.0c01313>
- 15 Sakamoto, K. M. *et al.* Development of PROTacs to target cancer-promoting proteins for ubiquitination and degradation. *Mol Cell Proteomics* **2**, 1350-1358 (2003). <https://doi.org/10.1074/mcp.T300009-MCP200>
- 16 Zhang, D., Baek, S. H., Ho, A. & Kim, K. Degradation of target protein in living cells by small-molecule proteolysis inducer. *Bioorg Med Chem Lett* **14**, 645-648 (2004). <https://doi.org/10.1016/j.bmcl.2003.11.042>
- 17 Inglese, J. *et al.* High-throughput screening assays for the identification of chemical probes. *Nat Chem Biol* **3**, 466-479 (2007). <https://doi.org/10.1038/nchembio.2007.17>
- 18 Cyrus, K. *et al.* Impact of linker length on the activity of PROTACs. *Mol Biosyst* **7**, 359-364 (2011). <https://doi.org/10.1039/c0mb00074d>
- 19 Rodriguez-Gonzalez, A. *et al.* Targeting steroid hormone receptors for ubiquitination and degradation in breast and prostate cancer. *Oncogene* **27**, 7201-7211 (2008). <https://doi.org/10.1038/onc.2008.320>
- 20 Qian, Y. C., A. P.; Dong, H.; Wang, J.; Crews, C. M. Indole derivatives as estrogen receptor degraders. World Intellectual Property Organization patent WO2018053354A1 (2018).
- 21 Crew, A. P. *et al.* Tetrahydronaphthalene and tetrahydroisoquinoline derivatives as estrogen receptor degraders. World Intellectual Property Organization patent WO2018102725A1 (2018).
- 22 Snyder, L. B. *et al.* Abstract 44: The discovery of ARV-471, an orally bioavailable estrogen receptor degrading PROTAC for the treatment of patients with breast cancer. *Cancer Research* **81**, 44-44 (2021). <https://doi.org/10.1158/1538-7445.Am2021-44>
- 23 Fan, J. & Liu, K. Novel compounds having estrogen receptor alpha degradation activity and uses thereof. United States patent US20180208590A1 (2018).
- 24 Neklesa, T. *et al.* in *American Society of Clinical Oncology (ASCO) Annual Meeting 2019*. Abstract 259 (American Society of Clinical Oncology).
- 25 Neklesa, T. *et al.* Abstract 5236: ARV-110: An androgen receptor PROTAC degrader for prostate cancer. *Cancer Research* **78**, 5236-5236 (2018). <https://doi.org/10.1158/1538-7445.Am2018-5236>
- 26 Rathkopf, D. E. *et al.* Safety and clinical activity of BMS-986365 (CC-94676), a dual androgen receptor ligand-directed degrader and antagonist, in heavily pretreated patients with metastatic castration-resistant prostate cancer. *Ann Oncol* **36**, 76-88 (2025). <https://doi.org/10.1016/j.annonc.2024.09.005>
- 27 Zeng, S. *et al.* Current advances and development strategies of orally bioavailable PROTACs. *Eur J Med Chem* **261**, 115793 (2023). <https://doi.org/10.1016/j.ejmech.2023.115793>

- 28 Scaturro, P. *et al.* Characterization of the mode of action of a potent dengue virus capsid inhibitor. *J Virol* **88**, 11540-11555 (2014). <https://doi.org/10.1128/JVI.01745-14>
- 29 Winter, G. E. *et al.* DRUG DEVELOPMENT. Phthalimide conjugation as a strategy for in vivo target protein degradation. *Science* **348**, 1376-1381 (2015).
<https://doi.org/10.1126/science.aab1433>
- 30 Donovan, K. A. *et al.* Mapping the Degradable Kinome Provides a Resource for Expedited Degradation Development. *Cell* **183**, 1714-1731 e1710 (2020).
<https://doi.org/10.1016/j.cell.2020.10.038>
- 31 Baek, K. *et al.* Unveiling the hidden interactome of CRBN molecular glues. *Nat Commun* **16**, 6831 (2025). <https://doi.org/10.1038/s41467-025-62099-w>
- 32 Jan, M., Sperling, A. S. & Ebert, B. L. Cancer therapies based on targeted protein degradation - lessons learned with lenalidomide. *Nat Rev Clin Oncol* **18**, 401-417 (2021).
<https://doi.org/10.1038/s41571-021-00479-z>
- 33 Uhlen, M. *et al.* Proteomics. Tissue-based map of the human proteome. *Science* **347**, 1260419 (2015). <https://doi.org/10.1126/science.1260419>

We have provided point-by-point responses (black text) to all Reviewer comments (blue text) below.

Reviewer #1 (Remarks to the Author):

1. The authors have satisfactorily addressed most of the questions and suggestions from the previous review, resulting in an improved manuscript.

Response: We thank Reviewer #1 for their assessment of the revised manuscript and for noting that the revisions have satisfactorily addressed the previous concerns.

2. For further strengthening, I would suggest performing a statistical analysis of the results shown in Supplementary Figure 8 (NanoBRET assay), as the observed change in BRET value appears modest.

Response: We performed statistical analysis of the NanoBRET data using one-way ANOVA with Dunnett's multiple comparisons test. Consistent with the reviewer's observation, BRET changes observed at a 1:10 donor-to-acceptor ratio are modest and do not consistently reach statistical significance after correction across independent experiments. Increasing the acceptor excess (1:100 donor-to-acceptor ratio) enhances the magnitude and consistency of the BRET signal, and dose-dependent increases reach statistical significance relative to DMSO after correction for multiple comparisons within the NanoBRET assay. Statistical analyses were performed within each independent experiment on technical replicates (n = 6–12 wells per condition) to assess assay-level reproducibility; accordingly, p-values reflect well-to-well variability and are presented as supportive evidence. Reproducibility of the direction and dose dependence of the response is demonstrated across independent experiments.

The following text is added to the Supplementary Figure 8 legend to include these statistical analyses: "Each data point represents a technical replicate (n = 6–12 wells per condition) within an independent experiment, and data are shown as mean ± SD. Statistical comparisons versus DMSO were performed within each experiment using one-way ANOVA with Dunnett's multiple comparisons test to assess assay-level reproducibility. At a 1:100 donor-to-acceptor ratio (replicate II), dose-dependent increases in BRET signal reach statistical significance relative to DMSO after correction for multiple comparisons (2.5 μM, adjusted p = 0.0020; 5 μM, adjusted p = 0.0004). Because these measurements represent technical replicates, p-values reflect well-to-well variability and are presented as supportive evidence. Replicates I and II represent two independent biological experiments."

3. Additionally, including a negative control—such as an irrelevant donor/acceptor pair—would help substantiate the conclusion.

Response: We thank the reviewer for this thoughtful suggestion. We agree that inclusion of an irrelevant donor/acceptor pair can be informative in NanoBRET assays. In the present study, however, the NanoBRET experiment is intended to serve as supportive evidence within a defined degrader assay framework rather than as a standalone demonstration of binding specificity. Assay performance and drug-dependent signal induction were benchmarked using a well-characterized degrader system (dBET6 with BRD4), and interpretation of the NanoBRET data is further supported by multiple orthogonal lines of evidence presented in the manuscript, including CRBN dependence, capsid degradation, and antiviral phenotypes in infected cells. In light of the supportive role of the NanoBRET data and the overall scope of the study, we have not added additional irrelevant donor/acceptor control pairs.

Reviewer #2 (Remarks to the Author):

1. The authors have done extensive new experiments to address all the concerns raised by three reviewers, including those points raised in my previous review.

Response: We thank Reviewer #2 for their careful re-evaluation of the revised manuscript and for their positive assessment of the additional experiments and overall quality of the study.

2. On balance, although RPG-01-132 is still a modestly potent PROTAC degrader of DENV capsid, the study nevertheless has provided high-quality of data to demonstrate the proof-of-concept for this compound and the overall direction. Importantly, the authors have included discussions to clearly state the limitations of RPG-01-132 and future directions. Therefore, I support the publication of this revised manuscript in *Nature Communications*.

Response: We appreciate Reviewer #2's recognition that the new data address the concerns raised in the previous review and that the manuscript clearly presents the proof-of-concept nature of RPG-01-132. We also thank the reviewer for acknowledging our discussion of the current limitations of RPG-01-132 and the articulation of future directions. We agree that while the degrader potency is modest, the work establishes a clear conceptual framework for capsid-directed targeted protein degradation as an antiviral strategy. We are grateful for the reviewer's support of publication in *Nature Communications*.

Reviewer #3 (Remarks to the Author):

1. In this revised version of their manuscript, the authors have substantially improved the study and addressed many of my important comments. They convincingly demonstrate that the mode-of-action of the drug is capsid-specific in infected cells and does not involve viral replication, translation or entry (to some extent in Fig S12).

Response: We thank Reviewer #3 for their thoughtful and constructive assessment of our revised manuscript. We appreciate their positive evaluation of the improvements made and their recognition that the antiviral activity of RPG-01-132 is capsid-specific and does not involve viral replication, translation, or entry. Below, we address each remaining comment point-by-point.

2. However, while replication and translation can be ruled out, the precise step targeted by the drug remains unclear. It is likely that assembly is targeted but this is not unambiguously shown as such.

Response: We agree with the reviewer that, while our data strongly support a capsid-dependent mechanism, the precise step within the late stages of the viral life cycle cannot yet be assigned unambiguously. Our electron microscopy data show a marked reduction in viral particle abundance under drug treatment, but we acknowledge that these observations are qualitative in nature and do not alone permit definitive assignment to a specific assembly step.

In response to this concern, we have revised the manuscript to avoid stating that virion assembly or formation is directly inhibited. Instead, in the abstract, we now describe the drug's effects as disrupting *capsid-related pathways required for productive infection*, which more accurately reflects the scope of the data and the reviewer's assessment. This revised framing preserves the central conclusion that capsid function is targeted while explicitly acknowledging current mechanistic limitations. In the discussion we have added the following revised text: "Although we cannot at this time distinguish a direct effect of RPG-01-132 on viral assembly from other potential mechanisms impacting capsid-dependent processes, the pharmacological effect of RPG-01-132 clearly differs from that of ST148." "We do not currently know if the degraders engage DENV C in a manner incompatible with ST148's mechanism or if degradation of DENV C occurs before it can be incorporated into a new virion." "Despite the challenges posed by the unique biology of viral infections, including spatial and temporal variability of protein expression, differential target abundance, and complex subcellular localization, we demonstrate that RPG-01-132 effectively degrades the capsid protein through an E3 ligase-dependent mechanism, resulting in disruption of capsid-related pathways required for productive infection, including reduced release of infectious virus and rescue of C-mediated antagonism of innate immunity."

3. Evidence of a decreased viral particle abundance is provided with electron microscopy data, but these observational phenotypes are not quantified. This is challenging to do. Thus, in order to support the conclusion, the authors should use the same experimental set-up as in Fig 2B, and measure in parallel intracellular and extracellular infectivity (by plaque assays) as well as viral intracellular and extracellular RNA (by RT-qPCR). If assembly is impaired, intracellular infectivity is expected to

decrease while intracellular RNA (at this 24 hpi time point) should remain constant (further validating the absence of impact on RNA synthesis/translation).

Response: We appreciate this valuable suggestion and agree that parallel measurements of intracellular and extracellular infectivity and RNA levels at 24 hpi would further refine mechanistic resolution. At present, our conclusions regarding the lack of effect on viral RNA synthesis and translation are based on early time-point measurements (2 and 8 hpi), which precede the onset of the antiviral phenotype and thus robustly exclude effects on early replication events.

Given the advanced stage of revision and the editors' guidance emphasizing the proof-of-concept nature of this work, we have not added these additional experiments in the current manuscript. Instead, we have explicitly acknowledged in the discussion that the precise late-stage step affected by RPG-01-132 remains to be determined and that future work will be required to distinguish between effects on assembly, maturation, or particle infectivity. The following text is added to the discussion: "C's multiple functions may be differentially affected by C degradation, with some capsid-related pathways more affected by depletion than others, both in terms of the susceptibility to the TPD mechanism due to differences in localization, binding partners, or other reasons but also in terms of the impact of reduced C on the viral pathway – for example, the threshold of depletion needed to impact C's role in assembly may differ from that needed to impact a function in signal transduction. Since our study monitored total intracellular C, additional experiments are needed to understand how the partial degradation phenotypes of RPG-01-132 and RPG-01-139 confer the potent, C-specific antiviral activity observed."

4. The authors have assessed viral RNA levels at 2 and 8 hours (Fig S12) but this should also be done at a time point at which the drug phenotype is observed, that is 24 hpi.

Response: We agree that measuring viral RNA at 24 hpi would provide complementary information. As noted above, we have clarified in the revised text that our RNA measurements at early time points establish that RPG-01-132 does not impair RNA synthesis or translation, while the later antiviral phenotype reflects disruption of downstream, capsid-dependent processes.

5. New Fig. S17 supports that the drug induces a decrease in capsid levels when expressed alone. However, the phenotype is rather modest (50% decrease at best for only one drug concentration) as compared to infection conditions (Fig 3) while it increased up to 10-fold at high concentration. Considering these mixed phenotypes, to unambiguously confirm that C is degraded upon drug treatment, similar experiment (at 0.62 μ M) should be repeated in MG132-treated cells or in CRBN KO cells, in which C levels should be rescued.

Response: We thank the reviewer for highlighting this point. We agree that capsid degradation is more pronounced in the context of infection than upon ectopic expression, which likely reflects differences in capsid abundance, localization, oligomerization state, or engagement with host factors during infection. Importantly, the infection-based experiments consistently show robust, CRBN-dependent loss of capsid concomitant with antiviral activity.

While additional rescue experiments using MG132 or CRBN knockout cells would further strengthen the mechanistic link, the current dataset already demonstrates CRL4^{CRBN} dependence through genetic and pharmacological approaches and establishes capsid as the relevant target in infected cells.

Minor comment: In the description of RT-qPCR assays, it is not appropriate to use the term capsid RNA which is in fact the viral (genomic) RNA. The way it is written implies that there are two different RNA species. There is only one viral protein-encoding RNA which is translating into the viral polyprotein. This must be corrected.

Response: We thank the reviewer for pointing out this imprecise terminology. We agree that the term "capsid RNA" is inappropriate and may misleadingly imply the existence of a distinct RNA species. In the revised manuscript, we have corrected this throughout the text to refer explicitly to *viral genomic RNA*, and we now clarify that the RT-qPCR assays quantify the single protein-encoding viral RNA that is translated into the viral polyprotein. We apologize for the confusion caused by the original wording.